# A classification scheme to determine wildfires from the satellite record in the cool grasslands of southern Canada: considerations for fire occurrence modelling and warning criteria

Dan K. Thompson[1], Kimberly Morrison[1]

[1]Canadian Forest Service, Northern Forestry Centre, Natural Resources Canada, Edmonton, Canada

*Correspondence to*: Dan K. Thompson (Daniel.Thompson@canada.ca)

**Abstract.** Daily polar orbiting satellite MODIS thermal detections since 2002 were used as the baseline for quantifying wildfire activity in the mixed grass and agricultural lands of southernmost central Canada.  This satellite thermal detection record includes both the responsible use of fire (e.g. for clearing crop residues, grassland ecosystem management, and traditional

burning), as well as wildfires in grasslands and agricultural lands that pose a risk to communities and other values.  A database of known wildfire evacuations and fires otherwise requiring suppression assistance from provincial forest fire agencies was used to train a model that classified satellite fire detections based on weather, seasonality, and other environmental conditions.  A separate dataset of high-resolution (Landsat 8 thermal anomalies) of responsible agricultural fire use (e.g. crop residue burning) was collected and used to train the classification model to the converse.  Key common attributes of wildfires in the

region included occurrence on or before the first week of May with high rates of grass curing, wind speeds over 30 km h$^{-1}$, relative humidity values typically below 40% and fires that are detected in the mid-afternoon or evening.  Overall, grassland wildfire is found to be restricted to a small number of days per year, allowing for the future development of public awareness and warning systems targeted to the identified subset of weather and phenological conditions.

## 1 Introduction

Wildfire is a widespread and commonplace phenomenon in Canada, with contexts ranging from an integral component of traditional land use (Lewis et al., 2018), a purely natural disturbance (i.e. lightning ignition) process with little human impact (Whitman et al., 2018), to a devastating natural hazard to communities (Christianson et al., 2019).  Fire (both human and natural ignition) is most common in Canada in its interior, west of the Great Lakes and east of the Rocky Mountains, where a belt of high fire frequency extends from the subarctic forests of the Deh Cho (Mackenzie Valley) through to the drier southern

boreal forest-grassland transition (Boulanger et al., 2014).  Within this broad north-south transect, the density of values at risk varies greatly, from sparse communities in the northern forest with limited industrial activities to a dense matrix of industry with dispersed agriculture and rural habitation (Johnston and Flannigan, 2018).  At the southern limit of the boreal forest in western Canada, climatic limitations to widespread forests created a natural ecotone towards a more open deciduous forest and grass parkland (Hogg, 1994; Zoltai, 1975), which has been almost entirely converted to intensive agriculture with a steady rate

of increasing agricultural conversion (Hobson et al., 2002). This is in contrast to the United States, where extensive natural grasslands intermix with dry conifer forests in areas of greater wildfire occurrence (Gartner et al., 2012). In Canada, at the southern forest limit and further south, the wildland-urban interface transitions to widespread human agriculture and only patches of broadleaf (deciduous) aspen forest (Hogg, 1994). Though smaller, localized grasslands in a larger matrix of forest are readily integrated into local wildfire likelihood assessments (Parisien et al., 2013), large-scale assessments of wildfire

likelihood are often based on modelling that utilizes forest fire management agency records (Parisien et al., 2013; Stockdale et al., 2019), and therefore exclude wildfires in agricultural areas where no such land management agency records exist. In this primarily agricultural region, controlled agricultural burning is commonly used to burn off excess crop residue (Chen et al., 2005). The use of a purely thermal remote sensing approach to determine the risk of wildfire (Rogers et al., 2015)  (i.e. fires being actively suppressed but not under control) is somewhat limited, and can erroneously count responsible fire use in

agriculture as wildfire occurrence.

In Canada, both forest fire and grass fire likelihood and spread are predicted using a common system, the Canadian Forest Fire Danger Rating System (CFFDRS), developed and maintained by the Canadian Forest Service starting in the 1930s. The system allows for the prediction of grass fire rate of spread (metres/minute), fire intensity (equivalent to flame height), and expected

growth rate (fire size over time). Fire weather is quantified using daily temperature, precipitation, humidity, and wind speed, with grass curing (the ratio of dead grass to live grass) being a critical variable that controls grass fire behaviour. Under the Canadian Fire Weather Index System (Van Wagner, 1987), the fire danger classes for public awareness (i.e., Low, High, Extreme, etc.) are based on a scaling of the expected head fire intensity of an idealized pine stand with a pine needle surface fuel bed. In this type of forest, wind speed, humidity, and drought will impact fire behaviour, but the lack of deciduous trees

or understory vegetation negate seasonal phenology beyond needle flush. When this Fire Weather Index scheme is then applied across regions dominated by grasslands, agriculture, or deciduous tree or shrubs, the Fire Weather Index alone and associated Fire Danger classes need to be adjusted for leaf-on or greenup conditions (Alexander, 2010; Chéret and Denux, 2011).

Recent research in Australia has highlighted the importance of grass fuel loading as a negative influence on fire rate of spread,

whereby a doubling of grass fuel load from the standard assumption of 0.35 kg of fuel m$^{-2}$ to 0.70 kg m$^{-2}$ results in a 10 % reduction in spread rate (Cruz et al., 2018). Conversely, a 50 % reduction in fuel load results in between a 10–30 % increase in spread rate; flame height (proportional to fireline intensity) increased to the power of 0.60 with increased fuel loading however, meaning a doubling of fuel loading results in a 50 % increase in flame height. Accordingly, under dry conditions, light agricultural residues may burn with high rates of spread though lower flame heights, while higher fuel loads in agricultural

residues would likely burn slower but with substantially larger flames. In mixed forest and open grass-type fuel landscapes, the lower intensity of grass fires typically results in higher rates of successful fire suppression for grasslands in empirical (Finney et al., 2009) and modelling (Reimer et al., 2019) studies compared to standing forest. Rapid fuel moisture gains during

typical night-time periods results in limited nocturnal fire activity potential (Kidnie and Wotton, 2015) except during exceptional periods of sustained wind and very low humidity (Lindley et al., 2019).


The overall goal of this study is to examine the differing environmental conditions most common during agricultural fires, and to contrast that with documented grassland wildfires in the region. The first specific goal is to apply a classification model to historical fire thermal detections (2002–2018) in order to determine the relative densities of agricultural burning and smaller, mostly undocumented grassland wildfires. The second goal is to develop an initial data-driven wildfire occurrence criteria

usable for public warning specific to grassland and agricultural regions of southern Canada.

## 2 Materials and Methods

### 2.1 Summary of datasets

MODIS thermal detections were used as a spatially unbiased record of fire activity in the study area. Each thermal detection was then associated with gridded data including grass curing (NDVI), as well as surface weather and fire weather variables

from the Canadian Fire Weather Index (FWI) System (Table 1). These thermal detections were then clustered and classified where possible into confirmed agricultural (Landsat 8, (Kato et al., 2018)) or wildfire using a fire occurrence database (Hanes et al., 2018) or evacuation records largely from media reports (Beverly and Bothwell, 2011). These known agricultural and wildfire hotspot clusters and associated fields were then used to create a Generalized Additive Model (GAM), which was used to classify the unknown hotspot clusters into agricultural or wildfires and produce maps of their relative occurrence (goal 1).

Additionally, a decision tree model was also built on the confirmed wildfire vs agricultural hotspot clusters, to provide simplified classification thresholds (goal 2) for use in fire operations and as the basis for potential public warning criteria.

### 2.2 Study Area

The study area encompasses the entire primary agriculture zone of central-western Canada (Prairies) as well as the forest-agriculture mix that extends north (to 58º N at its furthest point) and east to (as far as 96º W) where the shallow granitic soils

of the Canadian Shield are found (Fig. 1). The southern limit of the study area is the United States border at 49º N, and the western limit is the continuous forest and protected areas of the Rocky Mountains. The climate of the region is cool and continental, with mean annual temperature ranging from 0.6º C in Peace River to 5.9º C at Lethbridge. The number of frost-free days is as few as 119 in Peace River, and as many as 132 in areas east of Lethbridge. Foehn winds (locally known as Chinooks) on the eastern side of the Rocky Mountains cause periodic temperature increases above freezing during winter,

allowing for occasional winter fires in grass and other open, fine fuels. Snowmelt typically occurs in March-April in the southern extent, and April- early May further north. Annual precipitation varies from close to 600 mm in the easternmost edge of the study area near Winnipeg to as little as 316 mm in areas northeast of Lethbridge. Precipitation is heavily weighted to convective precipitation in the months of June-August. April and October are typically the two driest snow-free months.

Overall, 42 % of the study area is agricultural land or grasslands.  Land ownership in the agricultural area is almost entirely privately held, with the exception of First Nations reserves (1.6 %), parks and protected areas (2.4 %), and provincial grazing reserves (1.8 %).  Wildfire response is primarily volunteer-driven at the local community level (McGee et al., 2015).  At the fringe of agriculture, private land is intermixed with provincially (sub-national) owned lands that are managed primarily for timber, and wildfire response is entirely the responsibility of provincial fire management agencies outside of settlement boundaries.  Remotely-sensed land cover data at 30 m resolution (Agriculture and Agri-Food Canada, 2018)  was used to distinguish forested areas from open fuels (including permanent croplands, pastures, native grasslands, and treeless wetlands) all of which share similar phenology and flammability.  Broadleaf crops vs cereals were not distinguished.

**2.3 Fire occurrence records**

In the forest-agriculture mix, we used comprehensive fire history records from wildfire management agencies, as compiled in the Canadian National Fire Database (CNFDB) (Hanes et al., 2018). In the agricultural zone, the CNFDB provides only a partial sample of wildfires in the region.  The agricultural zone is not located in the provincial wildfire agencies' area of responsibility, therefore in this zone, only larger fires that required a mutual aid response from provincial agencies are documented in the database. Additional reporting on wildfire occurrence in the agricultural zone is provided by the Canadian Wildfire Evacuation Database (Beverly and Bothwell, 2011), which since 2010 has collected information on wildfire evacuation in grassland areas in addition to forest fires dating back to the 1980s.

Records from fire management agencies and evacuations provide a partial sample of the true extent of wildfires in the agricultural zone, and capture completely the occurrence of wildfire in the provincial forest.  Remotely-sensed thermal detection of active wildfire from the polar-orbiting NASA Aqua and Terra satellites that pass over Canada at nominally 13:30 local time (with a 01:30 overnight overpass) were used as a spatially unbiased (but time-limited) sample of fire activity in the area (Fig. 2).  Off-nadir collections (Freeborn et al., 2014) were also utilized and the detection-specific detection hour was used. A standard MODIS collection from 2002-2018 (MOD14A1 and MYD14A1) (Giglio, 2015) with 1 km resolution was screened for persistent industrial heat sources. These MODIS thermal detections were merged into hotspot clusters based on the detection's track and scan distance, in an attempt to group detections from the same fire together (see supplementary material).

A 3-km grid of daily basic surface meteorology at 12:00 (noon) local time (air temperature, humidity, 10-m wind speed, and precipitation sum over prior 24 h) as well as Canadian Fire Weather Index System variables using inverse-distance weighting (Lee et al., 2002) was constructed for every day during 2002-2018. The rasters constructed use the same surface station data as  McElhinny et al., 2020. The primary Fire Weather Index variables used include the Fine Fuel Moisture Code, Initial Spread Index, Duff Moisture Code, and Drought Code (Lawson and Armitage, 2008).  The Fine Fuel Moisture Code (FFMC) is a

model of moisture content for fine dead vegetation material at the forest floor of a closed-canopy forest. The FFMC utilizes all of the above basic surface meteorology to estimate drying rate with an exponential drying rate (time to loss of 2/3 of moisture content) of 18 hours. It is used here as a proxy for the moisture content of dense matted grass thatch, with relative

humidity alone a better proxy for the moisture content (Miller, 2019) and ignition capacity (Beverly and Wotton, 2007) of standing grass. High FFMC values indicate drier conditions, up to a maximum of 101. The Initial Spread Index (ISI) is the product of the FFMC and the square of wind speed and is proportional to the forward spread rate potential for grasslands and other open vegetated fuels (Hirsch, 1996). ISI is calculated daily and represents the peak potential rate of spread typically found in the later afternoon at the daily temperature maximum. The Duff Moisture Code (DMC) represents the moisture

content of a forest organic soil layer as estimated by a simple precipitation and evaporation model. It has an exponential drying rate of 12 days, and can be considered a metric of the bi-weekly soil moisture budget. Similarly, the Drought Code (DC) is a simple vertical water budget model (Miller, 2020) for a soil column with a 100 mm soil water capacity (similarly, larger values indicate drier conditions). In this manner, the DC has been shown to represent variations in surface water levels (Turner, 1972); a simple vertical water balance of precipitation and evaporation controls surface water extent in the prairies of Canada,

where water routing to streamflow and groundwater infiltration is limited (Woo and Rowsell, 1993). As such, the DC is a proxy for the extent of saturated soil areas (wetlands and other surface pond water) that when sufficiently dry, increase the continuity of fuels on the landscape.

We purposely utilized the longer-duration MODIS dataset from 2002 onwards, rather than the shorter duration VIIRS dataset

from 2012 onwards. Though both sensors are capable of fire detection in the mid-wave infrared, VIIRS is in theory capable of detecting smaller or less intense agricultural fires (Johnston et al., 2018; Zhang et al., 2017) which offers little advantage when the goal is the detection of larger wildfires in the region. Moreover, one of the goals of this study is to examine broad spatial trends in fire occurrence, where a longer record is ideal. Recently launched geostationary weather-oriented earth observation platforms such as GOES 16/17, Meteosat, and Himawari offer many advantages for monitoring short-lived

wildfires, with scan rates every 10–15 minutes (Hall et al., 2019). The northern latitude of the study area (49–59° N) causes a severe degradation of the pixel size of GOES geostationary fire detections to 4 km and limited capacity in accurately resolving fire radiative power (Hall et al., 2019). The dataset and classification criteria presented here can assist in improving the confidence in real-time wildfire detection in these areas with widespread intentional fire use in agriculture on the landscape.

All hotspot clusters with less than 40 % open fuels (grasslands, croplands, and treeless wetlands) were too influenced by fire behaviour in forests, and were excluded from the dataset. Within the agricultural ecumene, the vast majority of the region constitutes open fuels (Fig. 1), and little tree cover exists outside of shelter belt plantations which exist as single rows of trees (Piwowar et al., 2016). Area burned in forest-shrub-grass mixes typical of post-fire regeneration were eliminated from this study, as their suppression, land ownership, and vegetation ecology more closely mirror forests than grasslands (Whitman et

al., 2019). This resulted in a total of 24 297 MODIS hotspot clusters containing a total of 44 324 thermal detections. The

CNFDB and evacuation database were used to classify these hotspot clusters as wildfires where possible. Eighty-four hotspot clusters representing wildfires were identified using the CNFDB and 15 additional hotspot clusters were identified using the evacuation records and were not otherwise recorded in the CNFDB.

The responsible use of fire in the region includes traditional burning by First Nations (Lewis et al., 2018), prescribed burning by fire management agencies to reduce fuel loads in grasslands (McGee et al., 2015), burning of crop residues (Chen et al., 2005), and pile burning during land clearing operations where residual tree biomass is burned during agricultural land conversion (Hobson et al., 2002). Other than prescribed burning, no official documentation exists for this type of fire use, and could otherwise be conflated with wildfires as documented by remote sensing. In order to discriminate between responsible

fire use and wildfires, we used the 30 m short-wave infrared thermal detections from the Landsat 8 satellite (Kato et al., 2018) in order to classify clusters of thermal detections as responsible fire use if they correspond to geometric patterns associated with prescribed burning or other controlled fire (Fig. 3). A total of 41 Landsat hotspot clusters were manually classified in this manner; fire weather and land cover were associated with these detections similar to the MODIS detections. These Landsat detections are limited in spatial scale as the satellite only returns over an area every two weeks, so these records are at best a

small sample of the entire fire activity in the region (approximately 1/14, or 7 %), and only a small sample of Landsat data was used in this study. All responsible use of fire is referred to as agricultural fire in this paper.

### 2.4 Satellite grass curing

Grass curing (the fraction of dead grass with moisture content controlled by atmospheric conditions) is the primary control on the fire spread potential in grass fuels, overriding all other factors (Cruz et al., 2015). However, capturing the complexities of

plant phenology in the simple daily weather scheme used by the Fire Weather Index System or similar scheme is challenging (Jolly et al., 2005). For this retrospective analysis, we leverage satellite greenness as a proxy for grass curing, similar to (Pickell et al., 2017). In this study, we leverage historical 16-day composite NDVI (MOD13Q1 and MYD13Q1) (Didan et al., 2015) at 250 m resolution. A simple linear transform was used to convert between NDVI and percent curing:

$$P_{curing} = \left( 1 - \frac{NDVI_t - \min(NDVI)}{\max(NDVI) - \min(NDVI)} \right) \times 100 \tag{1}$$

Where $NDVI_t$ is the measured NDVI at time $t$, $min(NDVI)$ represents the per-pixel minimum snow-free NDVI value, and $max(NDVI)$ is the per-pixel maximum NDVI climatology. Both the min and max values are based on the average of the annual maxima and minima from 2002 to 2014 (*i.e.* $n = 12$ per pixel for both min and max calculations).

## 2.5 Classification of thermal detections

In total, 99 MODIS clusters (representing 386 total individual hotspots) were associated with documented wildfire, and 41 MODIS clusters (representing 104 total individual hotspots), confirmed to be agricultural controlled burning via Landsat imagery, were classified as agriculture fire use. Variables included for consideration in the GAM include surface weather variables, day of year, satellite curing fraction, as well as the fuel moisture codes (FFMC, DMC, DC) from the Fire Weather Index System. Higher-order components of the Fire Weather Index System such as Initial Spread Index and Buildup Index were not used due to their derivation from fuel moisture codes and high correlation (Spearman's $\rho > 0.7$) with those codes. The high correlation ($\rho = -0.73$) between relative humidity and FFMC is noted, but both were used in the GAM. All other variables in the GAM were correlated $\rho < 0.5$, and thus suitable for landscape-level fire weather analysis and modelling (Parisien et al., 2012). This data was then used to build models to classify the remaining hotspot clusters as either agriculture fire or wildfire using Generalized Additive Models (GAM) as binomial models (binary of wildfire or not) without interaction surfaces were built using the R package *mcgv* (Wood, 2019), with splines used for variables with an expected non-linear response such as ignition day of year, hour of detection (from MODIS), wind speed, and curing (Eq. 1). The non-linear partial effects terms in GAM models have been found to be superior to linear models with interactions in the examination of wildfire-environment data (Woolford et al., 2010). This model was validated using leave-one-out cross validation. These GAMs account for multiple non-linear responses but not interactions between predictors. Additionally, classification trees were constructed using the *rpart* package (Therneau et al., 2019) to classify wildfires from thermal detections using a simple conditional threshold-type model for use as simplified warning criteria (maximum of two variables). Inputs directly related to hotspot detection were not included (i.e. FRP), as they are only obtained upon fire detection. Variables that integrate multiple weather factors into a single index (i.e. Initial Spread Index or Buildup Index) were considered.

## 2.6 Analysis of classified clusters

The large dataset of hotspot clusters classified by the GAM were separated back into their individual hotspots (44 324), and used as a proxy for total fire on the landscape (a combination of fire size and fire occurrence). These classified hotspots were used to explore spatial and temporal patterns of agricultural and wildfires in the study area. The thresholds determined by both the decision tree model and GAM were used to produce a map representing the number of potential grassland wildfire days per year.

## 3. Results

Environmental, remotely sensed, and weather variables related to the distribution of agricultural vs wildfire hotspot clusters are shown in Fig. 4. Both fire types (agricultural vs wildfires) show a strong peak in the spring period after snow melt (Day of Year, Fig. 4a), centred on late April and early May, with a slightly earlier peak for wildfires. The curing fraction of the grass or agricultural residue is lower for wildfires compared to agricultural fires (Fig. 4b), which may be due to low NDVI(high

curing) artifacts from tillage (Zhang et al., 2018) or adjacent previously burned area in the larger MODIS pixels. The hour of first detection (Fig. 4c) is largely limited by the 13:00 local time overpass at nadir for MODIS. Night-time fire detections at 1am local (01:00) overpass are rare even for the wildfires. Pre-fire drying conditions as parameterized in the Fire Weather Index System (DMC and DC) show much larger right skews for wildfires. In the case of the DMC (Fig. 4d), 26% of the wildfire data have DMC values beyond the maximum DMC for agricultural burning of 67 (approx. 17 days without rain exceeding 1.5 mm). DC (Fig. 4e) shows a similar trend: 5% of agricultural fires have a DC of 470 or greater compared to 27% of wildfires. Observed fire weather values showed a meaningfully larger number of wildfires when relative humidity (Fig. 4f) was below 20 %, more agricultural fires when noon air temperatures are below 10°C (Fig. 4g), and far more wildfires when noon 10-m wind speeds exceed 25 km h$^{-1}$ (Fig. 4h). Fine Fuel Moisture Code (Fig. 4i) showed a peak for agricultural burning at FFMC 90 versus 92 for wildfires. Finally, the natural logarithm of the Fire Radiative Power (FRP) of the MODIS detection (Fig. 4j) showed far more variance in wildfires compared to agricultural fires. No agricultural fires exceeded 400 MW in the sample of confirmed agricultural fires. The median number of thermal detection points per wildfire was 2 but as high as 55, in contrast with agricultural fires where the median number of thermal detections is also 2 but the maximum is 6. Only 16 % of wildfires contained more than 6 hotspots in a cluster.

The above variables were assessed in a binomial generalized additive model, shown in Fig. 5. The GAM model was able to explain 64 % of the variance in the data, with strong non-linear predictors in Day of Year, curing, and wind speed. Day of Year analysis showed that wildfires are 75 % or more of detections for days prior to early May. Wind speeds over 25 km h$^{-1}$ or curing fractions between 50 and 85 % were also indicators of the likelihood of hotspot cluster being a wildfire. Relative humidity and DC were found to be significant in the GAM model as linear predictors, with odds ratios (increased likelihood of a fire being classified as a wildfire per integer increase in predictor value) of 0.31 per unit increase in RH, and 1.008 per unit of DC. Essentially, a single integer percentage increase in relative humidity, keeping all other measures constant, makes the odds of a hotspots detection being a wildfire drop by one third. Similarly, a 100 unit increase in DC (on a scale roughly from 0 to 700) makes the odds of a thermal detection being a wildfire increase by 80%. Despite the lack of interactions between predictors in all GAM models, the model had a high overall predictive power, with sensitivity of 0.86 (true positive rate), specificity (true negative rate) of 0.90, an area under receiver operating characteristic curve (AUC) of 0.89, and a Critical Success Index of 0.87 (Table 3). The cutoff of the overall GAM model binomial output of 0.66 provided the optimal model performance. When the GAM model is applied to the 24 297 hotspots clusters in the entire MODIS dataset, 30 % of hotspot clusters were detected under conditions that are most similar to documented wildfires. These hotspots have a strong regional gradient with more wildfires in the eastern portion of the study area (Fig. 6).

A simple decision tree was constructed from the same 140 classified hotspots to look at simple threshold-based classification schemes. (Fig. 7). An Initial Spread Index (ISI) (proportional to the fire's potential or modelled rate of spread based on weather alone) was found to be the strongest predictor, with 53 of 54 hotspot clusters being wildfires when ISI is greater than or equal

to 17. For fires with ISI <17, high curing (i.e. low NDVI, indicative of plowed fields in the vicinity or recent adjacent agricultural burning) over 87 % was a strong indicator of controlled agricultural burning, with 21 of 25 hotspot clusters being agricultural burning. Detection hour during or after 14:00 local time (indicating an intense fire detected in a later off-nadir satellite overpass) was also a meaningful indicator of a wildfire event, with 15 of 16 clusters being confirmed wildfires. For detections prior to 14:00 local time, an ISI of 11 or greater provided a moderately strong indicator of a wildfire, with 18 of 22 hotspot clusters detected being wildfires. For hotspots clusters with an ISI below 11, there was no meaningful discrimination between agricultural fires and wildfires. Overall, this decision tree model had an AUC of 0.75, and a favourable True Positive Rate of 0.77 with a lower True Negative Rate of 0.71 (Table 3). The Critical Success Index of this particular classification model 0.68 and overall Accuracy of 0.75.

The decision tree model was used to analyze the number of potential wildfire days per year given the criteria laid out in Fig 7. Geographic patterns of potential wildfire days (Fig. 8) is the opposite of observed densities of both agricultural and wildfire (Fig. 6), with more days conducive to wildfires in the west of the study area. The seasonal and spatial patterns along lines of equal longitude are portrayed in a Hovmoller time-longitude diagram in Fig. 9. For both agricultural and wildfires there is a concentration of fires in the spring (around weeks 17 to 21, late March to late April) and between longitudes 100-105° W. Agricultural fires have an additional concentration of hotspots in week 43 (late September).

In addition to the classification tree presented in Fig. 7, some properties of wildfires show meaningful breakpoints beyond which all agricultural fires values with or without meaningful differences to overall distribution (Fig. 4) or as a linear predictor in the GAM (Fig. 5). Median FRP between all agricultural burns (39 MW) and all wildfires (59 MW) are similar, and a non-parametric Mann-Whitney $U$ test on the two samples did not differ significantly (Mann–Whitney $U = 1860$, $n_1 = 113$, $n_2 = 41$, $p < 0.44$ two-tailed). However, on the higher end of FRP, wildfires showed a much larger right skew to the FRP values, with the 99th percentile of agricultural fire FRP of 233 MW, while this corresponded to the 86th percentile of wildfire FRP (or the largest 14% of the wildfire data). With the maximum observed wildfire FRP being 1174 MW, this allows for an additional logical scheme to discriminate wildfires from agricultural burning not captured in the above decision tree, where MODIS hotspot FRP values > 233 MW can be confidently classified as wildfires. Similarly, median noon wind speeds between agricultural fires (15.5 km h$^{-1}$) and wildfires (21.2 km h$^{-1}$) were similar, though distributions differed significantly (Mann–Whitney $U = 1387$, $n_1 = 113$ $n_2 = 41$, $p = 0.0001$ two-tailed). Some 30% of wildfire wind speeds exceeded the 90th percentile of agricultural fire wind speeds (22 km h$^{-1}$), allowing for an additional simple classification consideration for fire thermal detections during periods of high wind speed.

## 4. Discussion

In all likelihood, many of the roughly 7 500 wildfire hotspot clusters classified by the GAM over 17 years (or 441 fires per year over a 115 Mha study area) are smaller, briefly out of control fires where agricultural burning gets beyond direct suppression and burns over a number of adjacent agricultural fields until the wildfire encounters a roadway (typically over 10 m of fuel-free width), which readily stops most wildfires in grass and agricultural residue fuels (Cheney, and Sullivan, 2008). Given the generally widespread dispersed population density of the area, the vast majority of wildfires in the region are detected and reported by the public (McGee et al., 2015), such that satellites as the first mode of wildfire detection is of limited utility in the region, compared to more northerly and remote areas (Johnston et al., 2018). However, satellites provide a consistent technique for medium-resolution fire extent reporting and mapping that can prove useful for emergency managers (Lindley et al., 2019). Moreover, wildfire growth modelling (Sá et al., 2017) and smoke dispersion forecasts (Chen et al., 2019) require real-time analysis and forecasting initialized using remotely-sensed fire detections.

Both the GAM as well as the classification tree point to the combination of critically dry fuel and wind as the drivers of wildfire occurrence in the region. In the GAM model, both low RH (as a proxy for standing grass moisture), alongside indicators of bi-weekly (DMC) to monthly (DC) moisture deficit are significant in predicting wildfire occurrence as linear predictors, with a wind threshold in the range of 30 km h$^{-1}$. In the Canadian Fire Weather Index System, fine fuel moisture (mostly driven by low RH) is combined with wind speeds to calculate the Initial Spread Index as a single heuristic (Appendix A), and thus comes out as the strongest indicator of wildfire. Lindley et al., 2011 found no such moisture deficit as a driver of wildfire occurrence, and instead found that RH alone below 25 % and particularly below 20 % as responsible for most grassland wildfires in west Texas. In our study region, RH alone however is not an ideal proxy for fuel moisture across the wide range of air temperatures found in the region during wildfire, as RH alone does not account for variable vapour pressure deficit at different temperatures (Srock et al., 2018) that drives the equilibrium moisture content of standing grasses (Miller, 2019). Moreover, the extensive shallow water bodies in the region may contribute during periods of higher moisture surplus (i.e. low DMC and DC) to a fragmentation of fuel continuity, similar to the function of larger lakes to the north in Canada (Nielsen et al., 2016).

While more complex classification models with additional predictors were easily built using the *rpart* package, the goal in the classification tree model is to create a parsimonious model with simple application in short range (same day to 3 day outlook) guidance whether environmental conditions (grass curing, humidity, and wind speed) are sufficiently similar to historical wildfire occurrence. The classification tree presented in Fig. 7 is by no means the sole model that meets objectives for Critical Success Index and model accuracy. A high False Alarm Rate as present in the classification tree shown in Fig. 7 would be far more problematic in natural hazards such as tornadoes that require a sheltering response upon a false alarm (Ripberger et al., 2015). In this particular regional context, the criteria established to differentiate between agricultural fires and wildfires is more akin to the threshold beyond which responsible fire use activities should not occur due to dry and windy conditions,

rather than triggering a sheltering response. The adoption of any formal warning criteria requires a robust consultation process with regional stakeholders and is not within scope here. Rather, the data acquired and analyzed here provides for the efficient creation of future warning products in the region.

The study region is often impacted by prolonged dry periods. The study region experienced profound drought in the 1999–2005 period (Hanesiak et al., 2011) that corresponds to the start of the study period. Drought Code (representing a simple water balance of precipitation minus evaporation) was a weak linear ($p = 0.084$) predictor of grassland wildfire in this dataset, and may be considered mechanistically similar to the Palmer Drought Severity Index widely used in grassland and agricultural water availability studies (Hanesiak et al., 2011). Similarly, Duff Moisture Code as another weak linear indicator of wildfire detection, though a model as the drying of a forest floor organic soil, is still a metric of rainfall deficit relative to evaporation over the prior two weeks. Drought itself in the grasslands and agricultural areas of North America results in significant reductions in NDVI (Gu et al., 2007) that therefore directly increases grass curing as estimated in this study (Eq. (1)) and hence lengthens the seasonal window of grassland wildfire susceptibility.

The thresholds shown here in the classification tree and GAM models correspond to modelled fire intensity conditions at the upper limits of ground-based wildfire suppression. The grass fire spread model in the Canadian Forest Fire Danger Rating System utilizes Australian experimental grass fire data that has been shown to approximate fire behaviour in wheat crops, with the matted (or cut) grass model approximating spring (cured) post-harvest debris (Cruz et al., 2020). Following the Canadian Forest Fire Danger Rating System (Forestry Canada Fire Danger Rating Group, 1992) for an O-1a (matted grass) fuel type, ISI values of 17 (Fig. 7) with grass curing of 80 %, the resultant spread rate is 38 m min$^{-1}$ (2.3 km h$^{-1}$). This intensity of approximately 4000 kW m$^{-1}$ (flames 2 m long) is near the upper limit of suppression, particularly when fire sizes exceed 2 ha at the time of initial suppression action (Hirsch et al., 1998). This correspondence of our remotely sensed records (confirmed by fire reports solely of date and time, not of reported fire behaviour) and the operational models in the Canadian Forest Fire Danger Rating System lends confidence to the application of our approach in public safety and awareness messaging.

Under climate change, the agricultural and grassland region of Canada is anticipated to move northward (Schneider et al., 2009), though this rate of transition will be dampened in wetland areas (Schneider et al., 2016) and those not disturbed by wildfire (Stralberg et al., 2018). Natural grasslands are expected to increase particularly in areas of rapidly accelerating fire occurrence, where younger forests disturbed by severe wildfire are prone to large increases in grass cover (Whitman et al., 2019). Moreover, a dense grass cover is problematic in recently planted forests north of the study region, as it can outcompete tree seedlings (Lieffers et al., 1993), and is likely to be exacerbated by the expected lower overall canopy density (Lieffers and Stadt, 1994) brought about by a drier future climate (McDowell and Allen, 2015). Active conversion of forest to agricultural lands is likely to continue (Hobson et al., 2002), as is the natural expansion of grasslands on drier, south-facing (solar-exposed) slopes in the boreal forest (Sanborn, 2010). In addition to this likely grassland and cropland expansion, projections of

increasingly common critical fire weather conditions (Wang et al., 2015) is likely to shift the fire regime to one of more open fuel burning. However, no change in the rate of fire detections (undifferentiated between wildfires and agricultural burning) has been detected between 1981-2000 (Riaño et al., 2007) nor 1998-2015 (Andela et al., 2017) in the region.


The expansion of grasslands and agriculture into currently forested areas will substantially change the fire regime in these areas, highlighting the importance of understanding the current grassland and agriculture area fire regime. With grassland expansion into forest, forest fire suppression will have to incorporate elements of the grass fire regime. A key feature of the temporal nature of the grassland fire regime shown (Fig. 9) is the prevalence of wildfire in the month of April, far before

traditional forest fire suppression crews are trained and active (Tymstra et al., 2019). The occurrence of autumnal wildfire is much smaller than that in the spring (Fig. 9), but similarly requires resources for wildfire suppression in a region where the fire season has traditionally ended in early September (Hanes et al., 2018).

Regional contrasts in this grassland-agricultural fire landscape are revealed in the spatial analysis. Larger amounts of both

agricultural and wildfire in the east of the study area (Fig. 6), despite fewer burn days (Fig. 8), may be due to differences in agricultural practices or more flax agriculture. Farmers are more likely to burn flax crop residue as it can be difficult to remove by other methods (Chen et al., 2005). Higher rates of agricultural burning may also lead to increased wildfires via escaped fires from crop burning. Further work is required to better understand the contributions of vegetation and fire ignition that results in east-west gradients in both wildfire and agricultural activity observed here that contrasts directly with the number of

potential wildfire activity days.

## 5. Conclusions

A classification scheme was developed to discriminate remotely sensed agricultural fires vs wildfires in the southern grasslands of continental Canada through an analysis of historical wildfires and documented agricultural fires. Effective schemes for discriminating fire types were produced using continuous data (Generalized Additive Models) as well as threshold-based

classification trees. A combination of weather, vegetation condition, and temporal variables provided the best predictors. A noon Initial Spread Index threshold of >= 17 was the most powerful threshold from the decision tree model for discriminating wildfires from agricultural fires, while grass curing values between 60–85 % were the best non-linear spline predictor in the GAM. Fire Radiative Power was effective in discriminating wildfires only in the 14 % of wildfires with very high FRP values that exceeded the highest documented FRP in the agricultural fire dataset. Minor discrimination utility was seen in the Drought

Code and Duff Moisture Code precipitation deficit metrics. Classification of a large dataset of historical wildfire detections revealed a strong regional contrast in fire activity that is the inverse of the number of days with wildfire-conducive weather. Overall, the majority of the most power predictors of grassland wildfire stem from weather observations and remotely sensed

metrics of the pre-fire environment, and are thus available for forecasting and real-time classification of satellite thermal detections. This work provides a foundation from which future public warning products can be derived.

Author contributions: Conceptualization, DKT; Funding acquisition, DKT; Methodology, DKT and KM; Formal Analysis, DKT and KM; Data curation, KM; Investigation, KM; Project administration, DKT; Resources, DKT; Supervision, DKT; Validation, DKT and KM; Visualization, DKT and KM; Writing – original draft, DKT; Writing – review & editing, KM.

Funding: Funding for this project was provided by Crown Indigenous Relations and Northern Affairs Canada under the First Nations Adapt program.

Conflicts of Interest: The authors declare no conflicts of interest.

Acknowledgements: The authors would like to acknowledge Brett Moore and Chris Dallyn for providing valuable feedback on the manuscript and Peter Englefield for both feedback the acquisition of the MODIS hotspot and percent curing data.

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

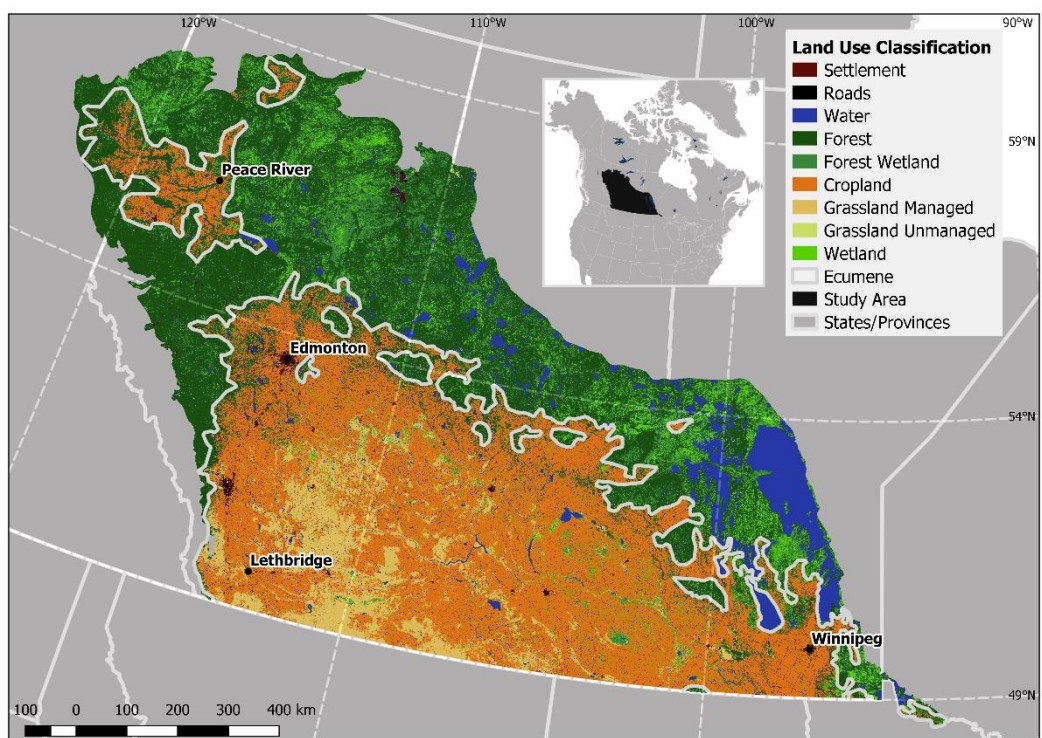

Figure 1. Remotely-sensed land cover data at 30 m resolution (Agriculture and Agri-Food Canada, 2018) of our study area as of 2010 compared to the extent of the ecumene. The study area extends past the ecumene to ensure all minor area of grass and agriculture are included.

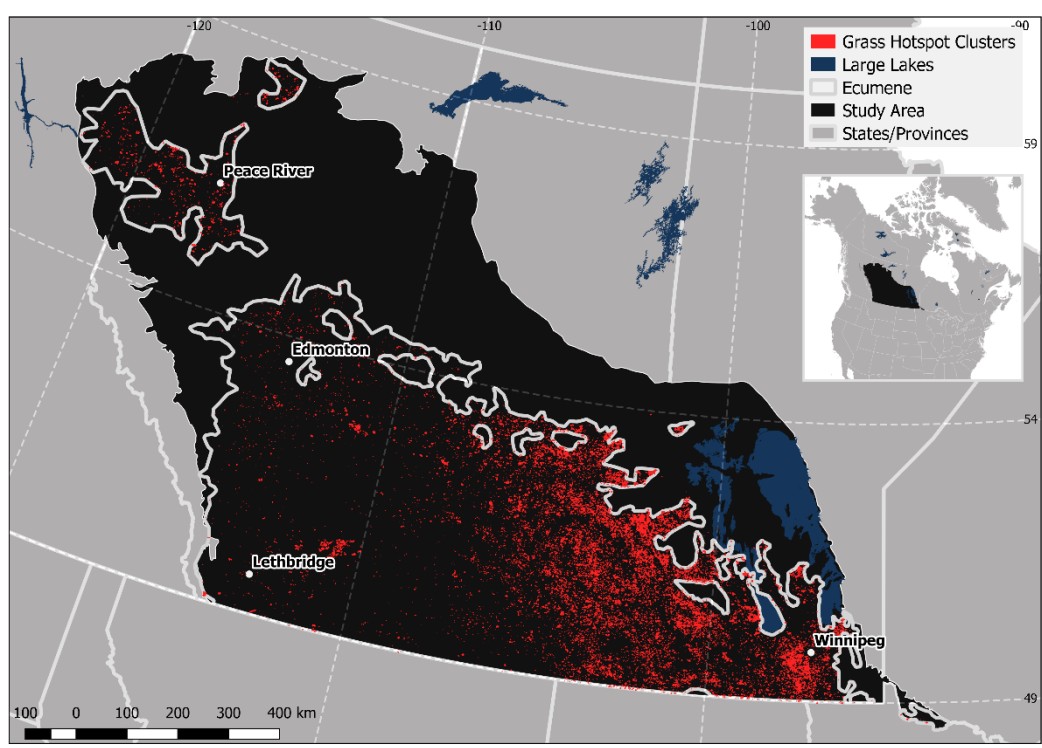

Figure 2: Grass fire MODIS(MOD14A1 and MYD14A1)(Giglio, 2015) hotspot clusters in the study area from 2002–2018. These hotspot have been screened for persistent industrial heat sources and clustered as described in the methods. The study area extends past the ecumene to ensure all minor areas of grass and agriculture are included. However, the vast majority of hotspot clusters are present within the ecumene.


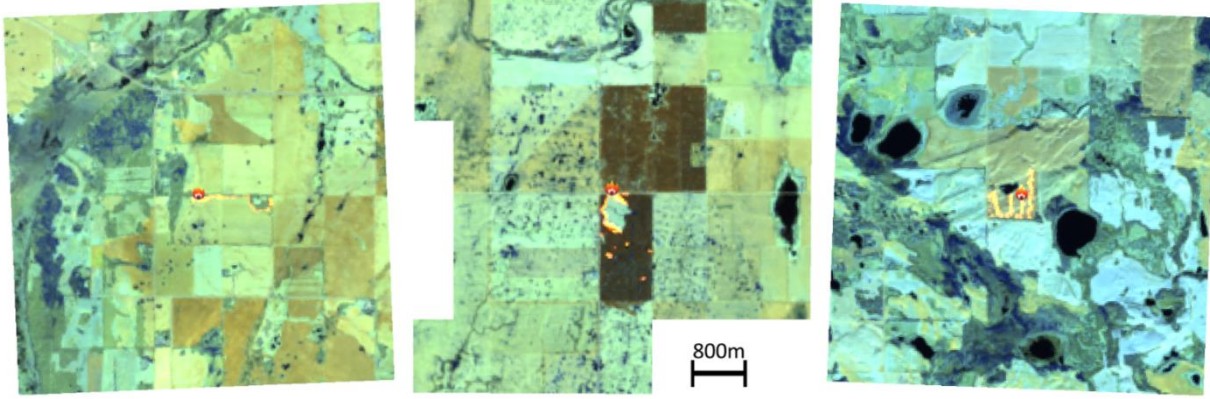

Figure 3. Examples of processed Landsat 8 images indicating fire detections considered agriculture burns. Note the regular geometric patterns of the fires, specifically the line ignitions patterns and the burning of specific fields. The presence of previously burned fields is shown north of the active fire in the centre panel, which is registered in this study as low NDVI and very high rates of curing.


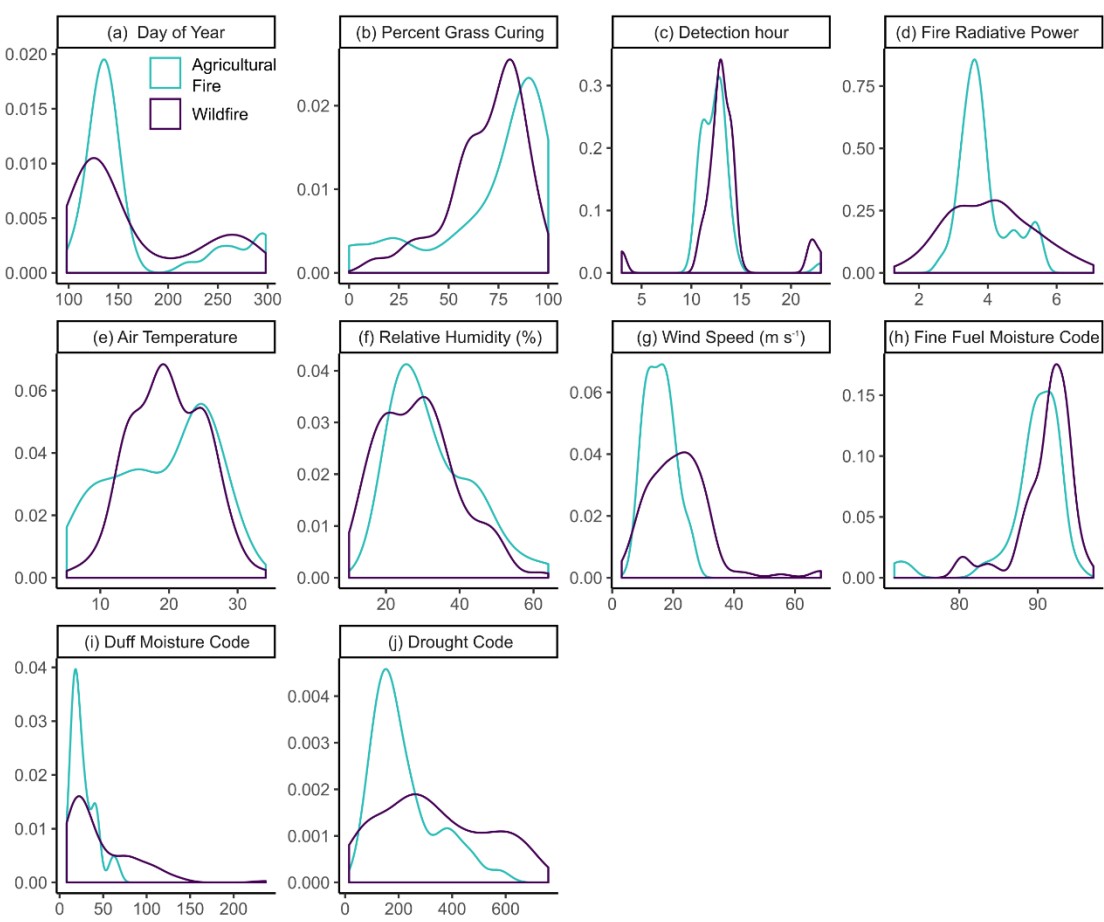

Figure 4. Distribution of hotspot cluster properties between wildfires (purple) and agricultural fires (blue). Fire Radiative Power is given in MW and transformed by the natural logarithm.

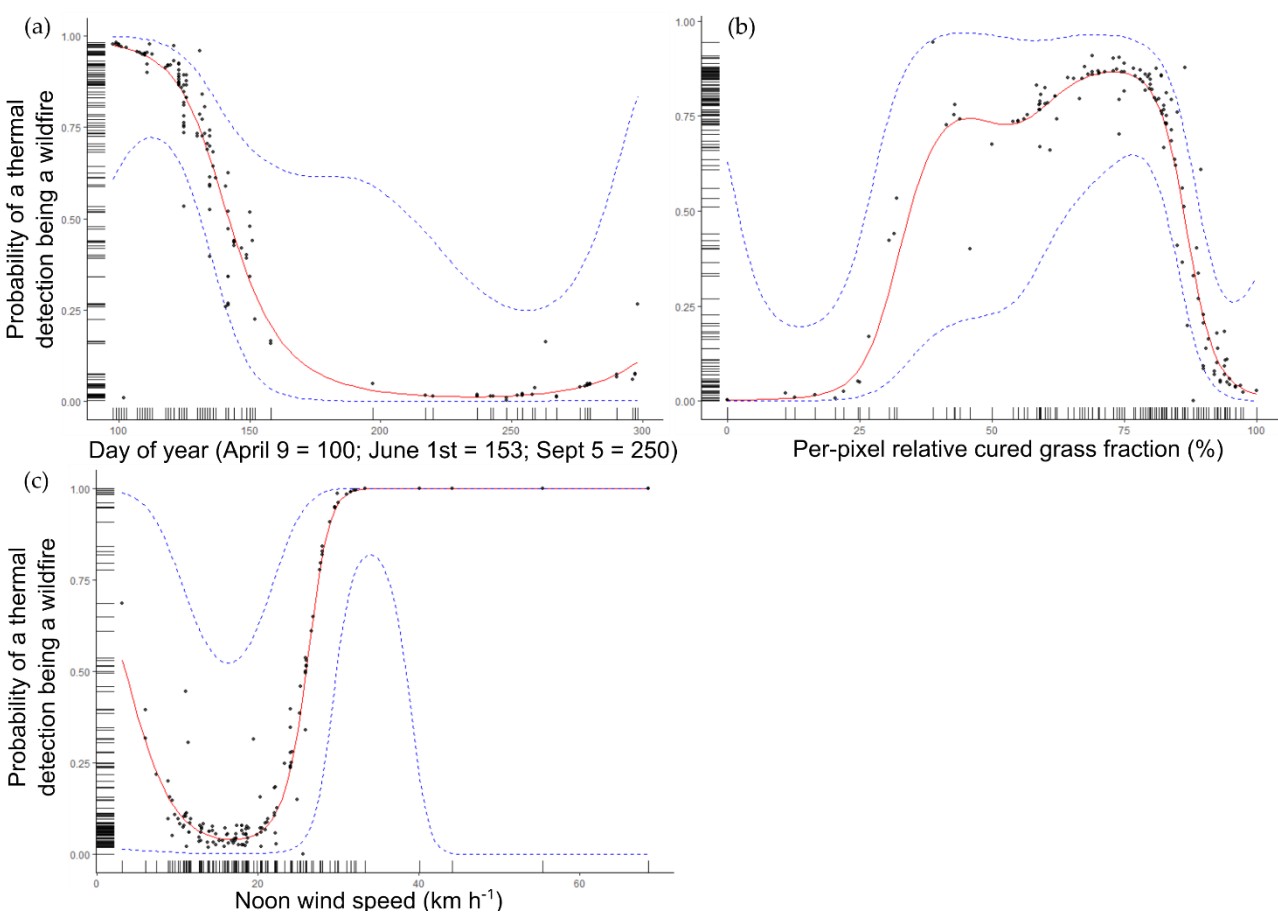

Figure 5: Generalized Additive Model outputs for a binomial model of agricultural fire vs wildfire. The ticks along each plot axis show the marginal distribution of the data.


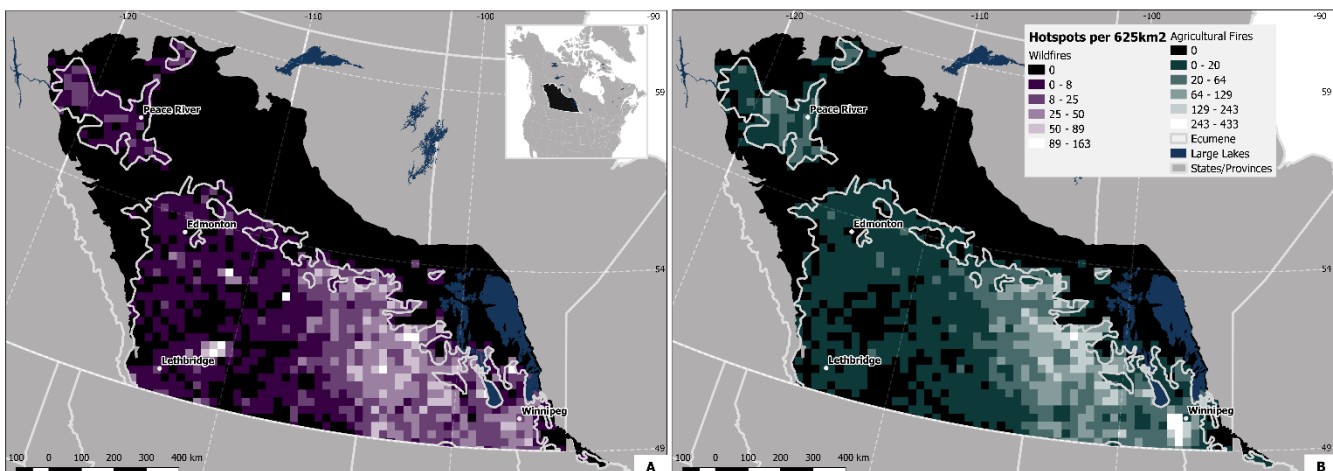

Figure 6. (a) Cumulative occurrence of wildfire hotspot detections per 25km square cell in the study region from 2002–2018. Panel (b) Cumulative occurrence of agricultural fire hotspot detections in the study region from 2002–2018. Discrimination between wildfire and agricultural fire hotspots conducted using the Generalized Additive Model (GAM).

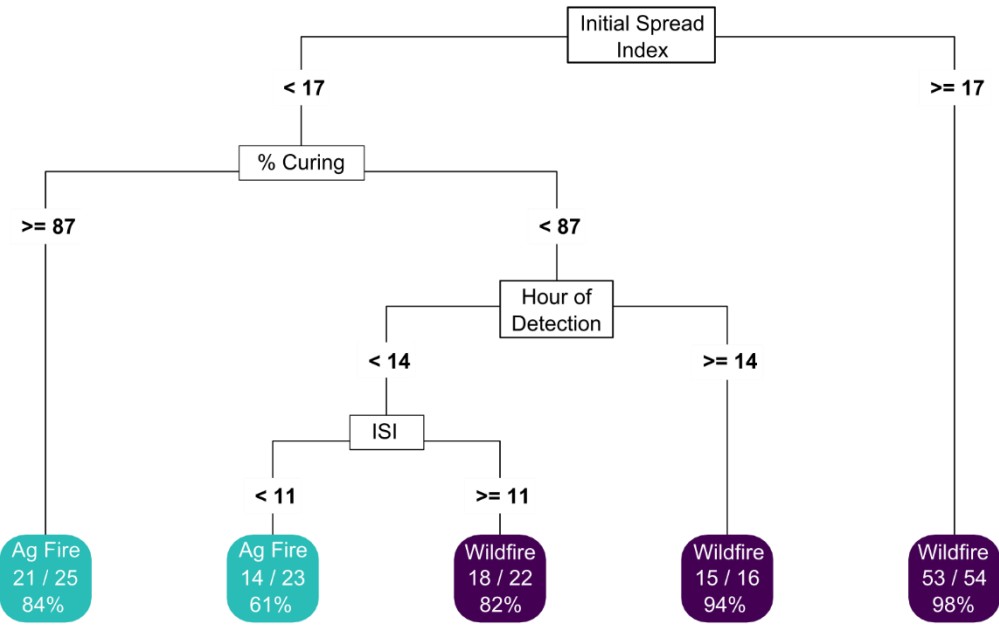

Figure 7. Simple decision tree scheme for the classification of agricultural vs wildfires. The first set of numbers in each terminal node is the number of correctly classified records divided by the total number of records in that node. The accuracy

of each node is also given. Note that high rates of curing is associated with plowed fields or those previously burned in agricultural fires in the days prior (see Fig. 5).


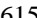

Figure 8. Average number of days per year (2002–2015, April–September) where the fire weather and environmental conditions meet or exceed an Initial Spread Index of 17 or greater as well as grass curing between 60 and 85 %.


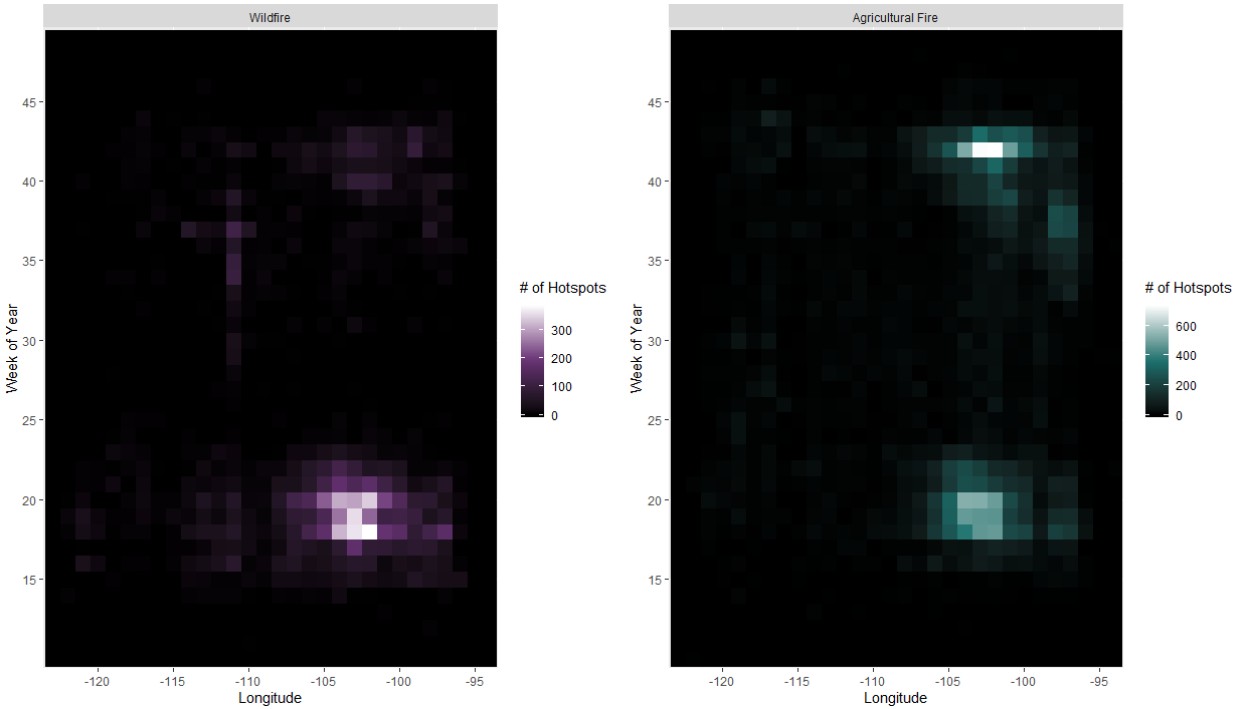

Figure 9. Hovmoller diagram showing seasonal patterns of wildfire vs agricultural fire. In this diagram, the number of hotspot detections is summed across all latitudes within a longitude bin (x-axis), and is shown over time (y-axis). Values are the cumulative sum of detections from 2002–2018.


Table 1: Summary of datasets used in study.

| Dataset | Spatial Resolution | Temporal Resolution | Derived Data | Product Number/Source | Time Frame |
|---------|---------|---------|---------|---------|---------|
| Land cover | 30 m | As of 2010 | Grass cover | (Agriculture and Agri-Food Canada, 2018) | 2010 |
| MODIS thermal detections | 1 km | Twice daily | Hotspot clusters | MOD14A1 and MYD14A1 | 2002-2018 |
| MODIS NDVI | 250 m | 16 day composite | Grass curing | MOD13Q1 and MYD13Q1 | 2002-2018 |
| Landsat 8 thermal detections | 30 m | 16 days | Confirmed agricultural fires | (Kato et al., 2018) | 2013-2018 |
| Weather and fire weather | 3 km grid | 12pm LST daily | Model input | (McElhinny et al., 2020)† | 2002-2018 |

| | | | Confirmed | (Hanes et al., | 2002- |
|---|---|---|---|---|---|
| Canadian National Fire Database | N/A | N/A | wildfires | 2018) | 2017 |
| Canadian Wildfire Evacuation Database | N/A | N/A | Confirmed wildfires | (Beverly and Bothwell, 2011)‡ | 2002- 2018 |

† The station data used in McElhinny et al (2020) were interpolated on a 3-km grid using an inverse distance weighting approach.

‡ The methodology of Beverly and Bothwell (2011) was applied to search for fires in the Prairie region of Canada, which were excluded from this publication. Evacuations were catalogued from 2002-2018. See supplementary data.


Table 2. The linear predictors of the GAM predicting if a hotspot cluster is a wildfire (coefficients given in logit space) alongside their odds for predictors with $p < 0.10$. Smooth of the GAM are shown with $X^2$ (Chi-square). [1]FRP = maximum Fire Radiative Power of a cluster (natural log-transformed); [2]RH = noon relative humidity (%), odds ratio per unit increase in
RH; [3]FFMC = Fine Fuel Moisture Code; [4]DMC = Duff Moisture Code.

| Linear Predictor | Estimate | SE | Odds Ratio | $p$ |
|---|---|---|---|---|
| Intercept | 14.5 | 11.6 | | 0.21 |
| Detection hour | 0.26 | 0.144 | 1.29 | 0.06 |
| [1]ln(FRP) | -0.30 | 0.30 | | 0.32 |
| Air Temperature | 0.054 | 0.078 | | 0.48 |
| [2]RH | -0.163 | 0.061 | 0.31 | 0.016 |
| [3]FFMC | -0.148 | 0.118 | | 0.21 |
| [4]DMC | 0.039 | 0.022 | 1.04 | 0.080 |
| Drought Code | 0.008 | 0.004 | 1.008 | 0.083 |
| Smooth Terms | *edf* | Ref *df* | $X^2$ | $p$ |
| Day of Year | 3.1 | 3.8 | 14.4 | 0.004 |
| Curing | 4.1 | 5.0 | 17.7 | 0.004 |
| Wind Speed | 2.0 | 2.1 | 9.4 | 0.034 |

Table 3. Generalized Additive Model (cutoff 0.66) and Decision Tree model performance metrics ($n = 140$ in both models).
Sensitivity, Specificity, and AUC (Area Under receiver operating characteristic Curve) were calculated using a leave-one-out cross validation. Miss Rate through to Accuracy Statistics were calculated using all data to train the model tested against itself.

| Metric | GAM | Decision Tree |
|---|---|---|
| True Positive Rate – Sensitivity | 0.87 | 0.77 |
| True Negative Rate – Specificity | 0.85 | 0.71 |
| AUC | 0.89 | 0.74 |
| False Negative Rate | 0.14 | 0.23 |
| False Positive Rate | 0.15 | 0.29 |
| False Discovery Rate | 0.06 | 0.13 |
| False Omission | 0.28 | 0.44 |
| Critical Success Index | 0.81 | 0.68 |
| Accuracy | 0.86 | 0.75 |

Appendix A. Observation density biplots of the fire weather associated with thermal detections (*n* = 3036) where the Initial Spread Index of the Canadian Fire Weather Index System is 15 or higher (3,036 of 24,316 total observations, or 12%). Panel A: noon vapour pressure deficit vs wind speed (both local noon standard time) for all observations of ISI 15 or higher. Panel B: relative humidity vs wind speed for the same subset. Panel C: Fine Fuel Moisture Code vs wind speed.

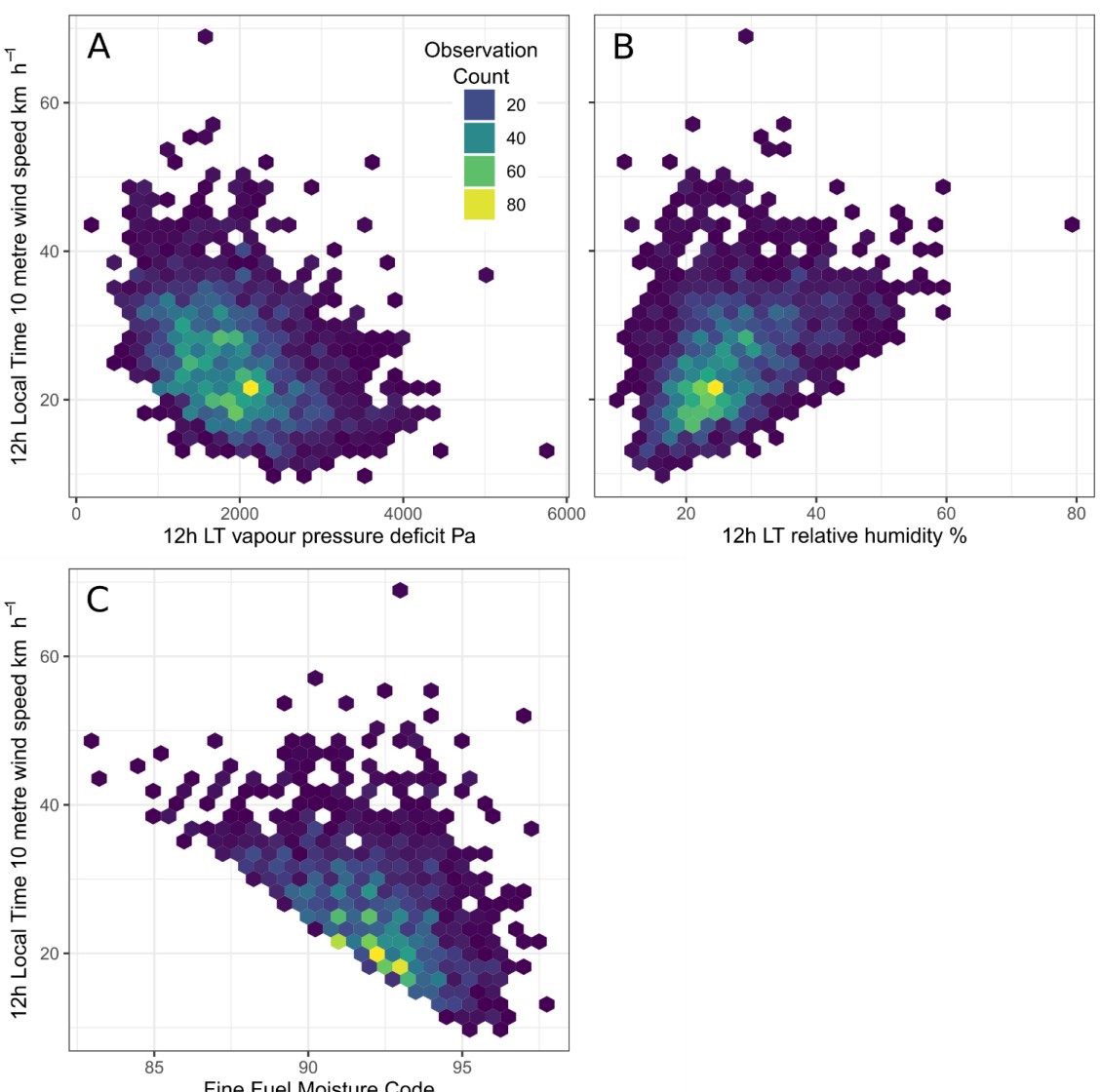