# Peer review of "A classification scheme to determine wildfires from the satellite record in the cool grasslands of southern Canada: considerations for fire occurrence modelling and warning criteria"

_Natural Hazards and Earth System Sciences, 2020_

## Referee Comment (RC1) · Anonymous Referee #1 · 22 Jun 2020

General comments

The authors present an interesting study that has practical implications for wildfire management in Canada and potentially beyond. The authors explore the discrimination of grassland wildfires from agricultural/managed) fires in South Central Canada. Using terrestrial datasets and high-resolution Landsat 8 data, the authors carefully construct and classify a dataset of MODIS fire clusters and explore the relationships between

these two classes of fire and various environmental/meteorological variables using GAMs and regression tree (RT) models. The work results in a series of parameter thresholds and value ranges that appear to be useful for pinpointing periods when wildfires are most likely, and could likely be used to enhance operational wildfire management in future.

This manuscript certainly merits publication in NHESS, however there are several areas where it could be improved prior to publication:

1) The narrative and structure could be improved throughout (see specific comments)

2) The methods need expanding, particularly with respect to the predictors chosen for inclusion in the models (some of this may be suited for inclusion in the supplementary materials).

3) Some of the results/discussion points could be elaborated on further, and the importance of this work better highlighted.

Specific comments

Abstract I would specifically refer to MODIS in the abstract, so it is immediately clear to readers what your primary RS dataset is.

1 Introduction

[39] For clarity I would amend to something like "...control) is somewhat limited, and can erroneously count responsible fire use..."

[44] also add precipitation here?

[65-70] It would be good to elaborate on the goals slightly here. From reviewing this paper, my understanding is that you are particularly trying to use the GAM for goal 1, and then goal 2 is informed by both models, so I would state this more explicitly. Also, related to this, I would probably address here (or perhaps in Section 2 somewhere, under the current structure possibly in 2.4) why you are building 2 different models e.g.

you use the tree approach primarily for explanation, and the GAM for both prediction & explanation? I don't think you clearly state anywhere your motivation for using 2 different approaches.

[66] I would say "documented grassland wildfires" here just to make the focus completely clear.

2 Materials & Methods General comment on methods: A lot of my questions / comments on Section 2 relate to dataset attributes that are either not provided or found in different subsections of the text. You use a lot of different datasets from different sources in this study, so a concise 'datasets' or 'materials' subsection that provides a list of each of these with useful basic information (data source, spatio/temporal coverage, resolution etc, as relevant) at the beginning of Section 2 would be useful – then readers can find all this information in one place without having to move backwards and forwards through different sub-sections.

2.1 Study Area

Figures 1 & 2: These figures are very nicely presented, however it is not immediately obvious which areas are the study area. At first, I thought it was all of the area for which landcover data are provided, i.e. including the forested regions. From looking at later figures, I think the Ecumene delineates the study area? I would suggest you consider changing the name of this to 'study area' for clarity, and change the colour of the boundary to something more obvious that the current grey, then maybe emphasise exactly where the study area is in the Figure 1 caption. Also, consider adding some of the info from the main body on the LC dataset to Figure 1 caption, and adding lines and/or ticks indicating lat/lon to these maps, as you refer to lat/lon locations in text - a reader not familiar with the region will have no context for this.

In figure 2 – What data is shown in this plot exactly? The full MODIS fire record (i.e. contains fires in non-grassland as well as grassland fires) or just data after the filtering step described on line 138? The label 'Grass hotspot clusters' – makes me think the

latter? As with Fig 1, I would expand this caption to include this kind of detail for clarity, also probably adding: whether this map is post-persistent hotspot filtering; the specific MODIS product it was derived from; the associated MODIS dataset citation.

2.2 Fire occurrence records

[93] I would consider stating the number of fires recorded over the study period from the CNFDB and evacuation dataset somewhere in this paragraph.

[94] I find the logic a bit confusing in the sentence on lines 94-96 (starting 'in the agricultural zone...') – perhaps reword it? You say that large fires are included in the CNFDB, but also that the CNFDB is 'only a partial sample' of fires in the region – so maybe you should be highlighting the fires that are not captured in this region by the CNFDB, rather than those which are?

[106] what is the source of the FWI data? Presumably the official CFFDRS datasets, but worth clarifying here.

[101-110] I think you should probably state the number of total MODIS fire detections and introduce the clustering concept here with detail on the number of fire 'clusters' resulting from the clustering process described in the supplement.

2.3 Satellite grass curing

[125] I'm confused as to how exactly this metric works. Eq 1 is the widely used min/max scaling (aka normalisation) applied to the NDVI climatology, so high values should imply high 'greenness' i.e. low curing? As such it is confusing to refer to this metric as 'percent curing' (also indicated in your GAM figure panel (b) x-axis as 'per-pixel relative cured grass fraction (%)'). Inverting this relationship (or renaming it) might be more intuitive? Moreover, from reading line [188] in the results you say "high percent curing (i.e. low NDVI..." which seems to be the opposite of Eq. (1), so there seems to be some confusion regarding how this metric is calculated somewhere? Did you actually invert this metric but omit this detail from this section?

I would also add the comment you make in the Landsat figure (the first Figure 3) caption to the main text somewhere here – that extremely high curing values can (somewhat counterintuitively) reflect prior agricultural burning/ploughing activity rather than dry veg – as this an important observation.

[138] I do not think the paragraph starting 'All hotspot clusters...' belongs in Section 2.3 as it stands. You don't really mention 'fire clusters' until Section 2.4, so it should go after this point. However, if you altered the MODIS paragraph [lines 100-110] in Section 2.2 by briefly describing the clustering process (that you describe in detail in the supp. materials), then this 'All hotspot clusters...' paragraph could follow fit in 2.2. Furthermore, it is probably worth explicitly stating how variables were aggregated by clusters, rather than your current explanation in the supplement "An attribute was merged by max value, min value or mean, for each hotspot cluster, whichever was most appropriate" [line 39], as this not very detailed.

2.4 Classification of thermal detections

[145] more background information regarding (1) the model predictors and (2) model construction process is definitely required in Section 2 (some of which could go in the appendices, if necessary). For clarity purposes, I think you definitely need to explicitly state and describe all the predictors that were added to the two models, along with their source (information could be in table form, and possibly a 'datasets' subsection as mentioned earlier), and where relevant, why those specific predictors were chosen over others. For example, FFMC, FWI, ISI often convey similar information -why was FFMC and not FWI or ISI chosen as predictors in the GAM, but ISI is used in the RT?

From reading the results section, I see that 'hour of detection' is derived from the MODIS dataset, however conceivably this could be information contained in the NFDB, and this sort of thing should be obvious from the methods.

I would indicate any standardisation / scaling of variables used in models here – e.g. DMC and DC were presumably scaled, as indicated in Fig 5(d) and detailed in section

3.

Did you test for and exclude any variables from the models based on collinearity using e.g. a simple correlation threshold? I assume you made some such decision here, as for example, you have omitted ISI & FWI as GLM predictors, and they are typically strongly correlated with e.g. FFMC. Similarly, I suspect RH and FFMC could be highly (negatively) correlated. Please explain how you addressed this. [149] what is your reasoning for not including interaction terms? Is this something that was initially explored and found to be unimportant, or were they not considered for simplicity reasons? I would be surprised if there were no relevant interactions between at least some of the predictors you have chosen to use.

[150] re: the argument for excluding curing from the RF model – does this logic not also extend to the GAM?

3 Results

[160-170] if you add a description of the predictor variables/datasets in Section 2, you can omit the 'background' info you include here: defining the DMC, explaining derivation of the FFMC, explaining the fact that FWI vars are observed at noon etc. These type of descriptions probably shouldn't appear in a results section.

[176] I would expand slightly here by highlighting what the significant splines show (I don't think you actually do this anywhere in the main body, but you do refer to the DoY criterion in the abstract?) e.g. wildfires are highly likely when: values of DoY < ~130, WS > 30, curing 65-85%.

Figure 5: You should explain panels (a)-(c) in the caption – at the minute you only mention panel (d). e.g. what are the blue lines (confidence intervals?) and black 'dashes' next to the axes (some kind of rug plot/distribution?).

Panel (d) of Figure 5 is a table, and so should be presented as such in the main body rather than as a panel of this figure. From Section 2.4 you suggest hour of detection

was incorporated as a spline not a linear predictor, but in (d) it is a linear predictor – which is correct?

Is there a reason why you didn't also include a plot of probability vs. FFMC in Figure 5 (as well as hour of day, if it was included as a spline?)

As mentioned earlier, DMC and DC are scaled before being used in the model, so this should be stated in the methods. Why does RH have an asterisk next to it? I would not use this symbol here as you already use asterisks to signify significance in the same table, which is confusing.

[184-201] decision tree results: This section is currently a bit confusing - I suggest it is restructured slightly, and some clarifications added. Firstly, how many fire clusters in total did you analyse here? I was expecting n=143 (113 wildfires + 41 agricultural fires stated on line 143, minus the 11 DC < 100 fires mentioned later) but adding up the denominators in Figure 6 it appears that n=95. Assuming I am reading Figure 6 correctly, shouldn't these two numbers match? After introducing the regression tree in Figure 6, It might be worth immediately stating the number of fires analysed, and that you removed the 11 low DC wildfires (plus any other filtering you did?) before discussing the specific results shown by the regression tree, so readers don't spend time looking for the 'missing' fires in figure 6.

[185] where is the 92 % accuracy figure from? Should this say 97 %? 92 % is not in Figure 6 or Table 1.

[186] Not sure why you talk about FFMC here – was FFMC actually used in the regression tree model? It doesn't appear in Figure 6.

[191] similarly, where does the 82 % value come from? not in Figure 6 or Table 1.

[192] I'm not sure about introducing Appendix A here, or actually including it in the paper at all (1) you don't really highlight what it adds to the study and (2) it uses the large fire dataset (>3000 fires) that you haven't really introduced yet.

[193-195] I would move the sentences comparing the GAM to the tree model, because you go from talking about just the tree model on line 192-3, to comparing the two models (193-195], and then back to discussing just the tree model [195-201], which is structurally hard to follow.

[203-215] this is interesting. Did you try including FRP & wind speed in the tree model? Seems like doing so could have added to tree classification skill?

[217-220] this paragraph (GAM applied to all clusters) feels like it might work better following the other paragraph on the GAM [lines 175-183]

[218] should this point to what is labelled as Figure 7 (the one with with two panel plots) rather than Figure 8?

Figures 7, 8 and 9: These are interesting figures, but you do not have much on them in either the results or discussion section (and in the case of figure 8, the 'avg. no. days per year figure', I don't think you mention this figure at all!). Some explanation is definitely required, otherwise why are they here?

4 Discussion

General comment on discussion:

Overall, you make some interesting points here, but several of them feel like they need expanding upon. I feel like you also don't draw much from the 'final' outputs of the study (Figures 7-9) – surely these results warrant discussion? Also, this paper clearly has important implications for operational fire management in grassland/agricultural complexes of Canada (and possibly beyond) – while you do mention this, I think you should try to highlight this aspect further in the discussion.

[222] this paragraph might go better in the introduction/datasets sections of the paper, as it is effectively a justification of why you chose to examine MODIS rather than other options

[232] "> 7500 fires" this statistic is from which dataset?

[244-252] I'm not sure what the key point you are trying to highlight here is, so this probably needs clarifying. I think your main point is that FFMC is a reasonable proxy for grassland moisture content/fire occurrence in the study area? If this is the case, it is interesting to me that (1) FFMC is not significant in the GAM and (2) FFMC is not included in the decision tree model (Fig 6), and this observation might merit further discussion here.

[261-266] Interesting observation, and this makes intuitive sense because managed fires that escape and become wildfires are probably usually the ones that reach the suppression limit. You should probably expand on this slightly though: (1) you could justifiably highlight that this adds to the validity of your work, as you have derived thresholds from a 'top down' RS/modelling approach that agree well with physical, bottom up observations of fire behaviour. (2) maybe you draw this out further? e.g. what might this finding have any applied fire management implications?

[268-276] You highlight an important point - that grasslands are increasing, and likely to keep doing so under climate change and current agricultural conversion trends. But you do not then use these points to highlight the importance of the work you have done here, and that it will be increasingly important in future – I think you should definitely emphasise this!

5 Conclusion

[283] maybe rephrase to say "a noon ISI threshold of > 15 was the most powerful threshold for discriminating wildfires from agricultural fires, while grass curing..."

Supplementary materials

[8] do you mean UTC rather than UTM date and time?

[39] How were each variable aggregated by fire cluster? E.g. FRP average vs max might be important to know...

[40] the 5% buffer you describe here, is this the same buffer you indicate in eq S2, or is this an additional buffer?

[117-122] this paragraph contains useful detail justification on the number of clusters you used. I would integrate at least some of this information into the main body, as it is important.

Technical corrections

Figures: Figure numbers are often incorrect in captions, and in places throughout the text. Please review and amend. Also consider generally expanding figure captions to include more information on the features of the figures or datasets used etc (see specific comments on figures where I believe these could be improved).

[60] consider deleting "….despite higher spread rates…". Probably adds to an unnecessarily long sentence

[87] "..northern fringe of agriculture.." - not sure if this applies to both areas (i.e. the 'main' southern Prairie area and the distinct northern agri-forest area?) or just the main southern one, please clarify

[96] "agencies" should have an apostrophe?

[Figure 4] is labelled as figure 2. You refer to panel letters (a, b etc) in the text but they are missing from the figure. I think this shows results for fire clusters, not MODIS pixel detections – make this clear in the caption and text. Also, the 'Day of year' panel only extends slightly beyond DoY 300 – is this intentional? (maybe there is never fire after this date?)

[150] is 'Additionally' a better word choice here than 'Alternately' as you build both models?

[155] I think you are referring to fire clusters here – if so, I would make this obvious by saying 'distribution of agricultural vs. wildfire clusters'

[173] I would state the median no. of pixels for agricultural fires here for comparison to the median wildfire pixels

[178] should this say "increased rate of wildfire likelihood per integer increase in predictor value"?

[257] I'm not very familiar with the use of odds ratios, so ignore this comment if it has a different technical interpretation - but might this be better phrased as "..results in the increase in the odds of a wildfire over an agricultural fire by 2.45. . ."?

[273] I think you want 'exacerbated' rather than 'exasperated' here?
* * *

---

## Referee Comment (RC2) · Jessica McCarty (Referee) · 2 Jul 2020

Overall, I think this paper is trying to advance natural hazards - specifically fire science - in using remote sensing and data science to attribute and predict wildland vs. human-caused fire. I would recommend the authors refine the terminology. I look forward to reading a revised version.

General comments: 1. Landsat 8 is not an acronym and should not be capitalised.

[Figure]

2. Referring to all non-wildland fires as agricultural fires becomes confusing later on, especially when trying to explain how the curing data set was included in the regression tree [much of the agricultural landscape was exempted from the curing assessment because < 40% open fuels].

The term 'responsible use of fire' is used to encompass a large amount of human-caused burning. Is this a legal or statute-based definition? This is not a common term in fire science. Also, burning of crop residues is not necessarily considered an appropriate thing for this ecosystem. The Province of Alberta has shifted to no-burn management of crop residues, treating burning as a last resort: https://open.alberta.ca/dataset/dd5ca66a-09f6-4aeb-8bb9-21babed92780/resource/3b67de8e-7377-406c-94d7-25f3efaee710/download/mar-2017-unharvested-crops-fs.pdf

Why was 2002 (Terra only) MODIS active fire product included when the combined (Aqua and Terra) MODIS active fire product is available starting in 2003? How were these differences in number of detections accounted for when determining the clusters? Was the 2002 Terra-only MODIS active fire useful?

Paragraph 265: The thesis statement of this paragraph may need to be re-written " The thresholds at which agricultural fire detections are overtaken by wildfires occurs at fire intensity thresholds that correspond to the limits of ground-based wildfire suppression." Is this a result or a qualitative observation or an assumption that fits into the description of the CFFDRS is the following sentence? Please consider re-phrasing this paragraph. I do not understand how this fits into the study or the findings. Perhaps, again, it is an issue with referring to grass fires as agricultural fires. This reads as the CFFDRS for native grasslands. Is that correct?

Is the last paragraph in the discussion section implying increasing agricultural fires with climate change? Did this study find increasing agricultural fires? And if so, in grasslands or croplands?

---

## Author Comment (AC1) · 25 Aug 2020

Reviewer 1 General comments The authors present an interesting study that has practical implications for wildfire management in Canada and potentially beyond. The authors explore the discrimination of grassland wildfires from agricultural/managed) fires in South Central Canada. Using terrestrial datasets and high-resolution Landsat 8 data, the authors carefully construct and classify a dataset of MODIS fire clusters and

explore the relationships between these two classes of fire and various environmental/meteorological variables using GAMs and regression tree (RT) models. The work results in a series of parameter thresholds and value ranges that appear to be useful for pinpointing periods when wildfires are most likely, and could likely be used to enhance operational wildfire management in future. This manuscript certainly merits publication in NHESS, however there are several areas where it could be improved prior to publication: The narrative and structure could be improved throughout (see specific comments) The methods need expanding, particularly with respect to the predictors chosen for inclusion in the models (some of this may be suited for inclusion in the supplementary materials). 3) Some of the results/discussion points could be elaborated on further, and the importance of this work better highlighted. »> We thank the reviewers for their careful reading of the manuscript, and we document our responses and revisions below.

Specific comments Abstract I would specifically refer to MODIS in the abstract, so it is immediately clear to readers what your primary RS dataset is. »>We revised the abstract to read: " Daily polar orbiting satellite MODIS thermal detections since 2002 were used as the baseline for quantifying wildfire. . ."

1 Introduction

[39] For clarity I would amend to something like ". . .control) is somewhat limited, and can erroneously count responsible fire use. . ." »> Revised as suggested.

[44] also add precipitation here? »>Changed to : "Fire weather is quantified using daily temperature, precipitation, humidity, and wind speed. . ."

[65-70] It would be good to elaborate on the goals slightly here. From reviewing this paper, my understanding is that you are particularly trying to use the GAM for goal 1, and then goal 2 is informed by both models, so I would state this more explicitly. Also, related to this, I would probably address here (or perhaps in Section 2 somewhere, under the current structure possibly in 2.4) why you are building 2 different models e.g.

[Figure]

you use the tree approach primarily for explanation, and the GAM for both prediction & explanation? I don't think you clearly state anywhere your motivation for using 2 different approaches. »> We've added a whole section 2.1 that immediately follows the goals stated at the end of the introduction. The section more explicitly states the workflow. We explain the rationale for the simplified decision tree at the end of section 2.1: "These known agricultural and wildfire hotspot clusters and associated fields were then used to create a Generalized Additive Model (GAM), which was used to classify the unknown hotspot clusters into agricultural or wildfires and produce maps of their relative occurrence (goal 1). Additionally, a decision tree model was also built on the confirmed wildfire vs agricultural hotspot clusters, to provide simplified classification thresholds (goal 2) for use in fire operations and as the basis for potential public warning criteria."

[66] I would say "documented grassland wildfires" here just to make the focus completely clear. »>Changed to: "...to contrast that with documented grassland wildfires in the region."

2 Materials & Methods General comment on methods: A lot of my questions / comments on Section 2 relate to dataset attributes that are either not provided or found in different subsections of the text. You use a lot of different datasets from different sources in this study, so a concise 'datasets' or 'materials' subsection that provides a list of each of these with useful basic information (data source, spatio/temporal coverage, resolution etc, as relevant) at the beginning of Section 2 would be useful – then readers can find all this information in one place without having to move backwards and forwards through different sub-sections. »> We now provide Table 1, which summarizes the datasets and their properties. We thank the reviewers for this helpful suggestion.

2.1 Study Area Figures 1 & 2: These figures are very nicely presented, however it is not immediately obvious which areas are the study area. At first, I thought it was all of the area for which landcover data are provided, i.e. including the forested regions. From looking at later figures, I think the Ecumene delineates the study area? I would

suggest you consider changing the name of this to 'study area' for clarity, and change the colour of the boundary to something more obvious that the current grey, then maybe emphasise exactly where the study area is in the Figure 1 caption. Also, consider adding some of the info from the main body on the LC dataset to Figure 1 caption, and adding lines and/or ticks indicating lat/lon to these maps, as you refer to lat/lon locations in text – a reader not familiar with the region will have no context for this. »> Map insets have been modified to show only study area, all in black to help clarify study area. Lat/lon grid lines were added to maps. Caption revised to: "Remotely-sensed land cover data at 30 m resolution (Agriculture and Agri-Food Canada, 2018) of our study area as of 2010 compared to the extent of the ecumene. The study area extends past the ecumene to ensure all grass and agriculture are included. The majority of fires analyzed in this study occurred within the agricultural ecumene."

In figure 2 – What data is shown in this plot exactly? The full MODIS fire record (i.e. contains fires in non-grassland as well as grassland fires) or just data after the filtering step described on line 138? The label 'Grass hotspot clusters' – makes me think the latter? As with Fig 1, I would expand this caption to include this kind of detail for clarity, also probably adding: whether this map is post-persistent hotspot filtering; the specific MODIS product it was derived from; the associated MODIS dataset citation. »>Caption changed to: "Figure 2: Grass fire MODIS(MOD14A1 and MYD14A1)(Giglio, 2015) hotspot clusters in the study area from 2002–2018. These hotspot have been screened for persistent industrial heat sources and clustered as described in the methods."

2.2 Fire occurrence records [93] I would consider stating the number of fires recorded over the study period from the CNFDB and evacuation dataset somewhere in this paragraph. »>Added: ". Eighty-four hotspot clusters representing wildfires were identified using the CNFDB and 15 additional hotspot clusters were identified using the evacuation record and were not otherwise recorded in the CNFDB." Added after clustering description.

[94] I find the logic a bit confusing in the sentence on lines 94-96 (starting 'in the

agricultural zone...') – perhaps reword it? You say that large fires are included in the CNFDB, but also that the CNFDB is 'only a partial sample' of fires in the region – so maybe you should be highlighting the fires that are not captured in this region by the CNFDB, rather than those which are? »> Revised to: " In the agricultural zone, the CNFDB provides only a partial sample of wildfires in the region. The agricultural zone is not located in the provincial wildfire agencies area of responsibility, therefore, in this zone, only larger fires that required a mutual aid response from provincial agencies are documented in the database."

[106] what is the source of the FWI data? Presumably the official CFFDRS datasets, but worth clarifying here. »> We clarify the source of the FWI data as: "A 3-km grid of daily basic surface meteorology at 12:00 local time (air temperature, humidity, 10-m wind speed, and precipitation sum over prior 24 h) as well as Canadian Fire Weather Index system variables using inverse-distance weighting (Lee et al., 2002) was constructed. The rasters constructed use the same surface station data as McElhinny et al., 2020."

[101-110] I think you should probably state the number of total MODIS fire detections and introduce the clustering concept here with detail on the number of fire 'clusters' resulting from the clustering process described in the supplement. »> Revised as suggested. We rearranged the manuscript so the clustering process was introduced here and included the number of clusters.

2.3 Satellite grass curing [125] I'm confused as to how exactly this metric works. Eq 1 is the widely used min/max scaling (aka normalisation) applied to the NDVI climatology, so high values should imply high 'greenness' i.e. low curing? As such it is confusing to refer to this metric as 'percent curing' (also indicated in your GAM figure panel (b) x-axis as 'per-pixel relative cured grass fraction (%)'). Inverting this relationship (or renaming it) might be more intuitive? Moreover, from reading line [188] in the results you say "high percent curing (i.e. low NDVI..." which seems to be the opposite of Eq. (1), so there seems to be some confusion regarding how this metric is calculated somewhere? Did you actually invert this metric but omit this detail from this section? »> Our mistake. The equations should have a 1 minus in front of it. That is a typographical error, and the equation in the analysis is correct. Low NDVI is high curing, and high curing corresponds to high fire potential. We use this curing fraction value to be consistent with the fire behaviour models, which using curing % from 0 to 100, rather than NDVI or greenness.

I would also add the comment you make in the Landsat figure (the first Figure 3) caption to the main text somewhere here – that extremely high curing values can (somewhat counterintuitively) reflect prior agricultural burning/ploughing activity rather than dry veg – as this an important observation. »> In the first paragraph of the results, we now state: "The curing fraction of the grass or agricultural residue is lower for wildfires compared to agricultural fires (Fig. 4b), which may be due to low NDVI(high curing) artifacts from tillage (Zhang et al., 2018) or adjacent previously burned area in the larger MODIS pixels. " We did not before add the comment on the previously burned area as we did in the figure caption.

[138] I do not think the paragraph starting 'All hotspot clusters...' belongs in Section 2.3 as it stands. You don't really mention 'fire clusters' until Section 2.4, so it should go after this point. However, if you altered the MODIS paragraph [lines 100-110] in Section 2.2 by briefly describing the clustering process (that you describe in detail in the supp. materials), then this 'All hotspot clusters...' paragraph could follow fit in 2.2. Furthermore, it is probably worth explicitly stating how variables were aggregated by clusters, rather than your current explanation in the supplement "An attribute was merged by max value, min value or mean, for each hotspot cluster, whichever was most appropriate" [line 39], as this not very detailed. »> Rearranged as suggested. Revised to include how variables were merged in supplementary material.

2.4 Classification of thermal detections [145] more background information regarding (1) the model predictors and (2) model construction process is definitely required in Section 2 (some of which could go in the appendices, if necessary). For clarity purposes, I think you definitely need to explicitly state and describe all the predictors that were added to the two models, along with their source (information could be in table form, and possibly a 'datasets' subsection as mentioned earlier), and where relevant, why those specific predictors were chosen over others. For example, FFMC, FWI, ISI often convey similar information -why was FFMC and not FWI or ISI chosen as predictors in the GAM, but ISI is used in the RT? »> We now clarify our choice in model selection inputs. For the GAM: "Variables included for consideration in the GAM include surface weather variables, day of year, satellite curing fraction, as well as the fuel moisture codes (FFMC, DMC, DC) from the Fire Weather Index system. Higher-order components of the Fire Weather Index System such as Initial Spread Index and Buildup Index were not used due to their derivation from fuel moisture codes and high correlation (Spearman's > 0.7) with those codes." And for the classification tree, we clarify: "Additionally, classification trees were constructed using the rpart package (Therneau et al., 2019) to classify wildfires from thermal detections using a simple conditional threshold-type model for use as simplified warning criteria (maximum of two variables). Inputs directly related to hotspot detection were not included (i.e. FRP), as they are only obtained upon fire detection. Variables that integrate multiple weather factors into a single index (i.e. Initial Spread Index or Buildup Index) were considered."

From reading the results section, I see that 'hour of detection' is derived from the MODIS dataset, however conceivably this could be information contained in the NFDB, and this sort of thing should be obvious from the methods. I would indicate any standardisation / scaling of variables used in models here – e.g. DMC and DC were presumably scaled, as indicated in Fig 5(d) and detailed in section 3. » We now clarify that the NFDB does not contain hour of detection data. We now clarify in section 2.2: "Off-nadir collections (Freeborn et al., 2014) were also utilized and the detection-specific detection hour was used."

Did you test for and exclude any variables from the models based on collinearity using e.g. a simple correlation threshold? I assume you made some such decision here,

as for example, you have omitted ISI & FWI as GLM predictors, and they are typically strongly correlated with e.g. FFMC. Similarly, I suspect RH and FFMC could be highly (negatively) correlated. Please explain how you addressed this. »> We now clarify this in the methods: "Higher-order components of the Fire Weather Index System such as Initial Spread Index and Buildup Index were not used due to their derivation from fuel moisture codes and high correlation (Spearman's > 0.7) with those codes. The high correlation ( = -0.73) between relative humidity and FFMC is noted, but both were used in the GAM. All other variables in the GAM were correlated < 0.5, and thus suitable for landscape-level fire weather analysis and modelling (Parisien et al., 2012)."

[149] what is your reasoning for not including interaction terms? Is this something that was initially explored and found to be unimportant, or were they not considered for simplicity reasons? I would be surprised if there were no relevant interactions between at least some of the predictors you have chosen to use.] »> We've added the text to clarify our choice in GAM models: " The non-linear partial effects terms in GAM models have been found to be superior to linear models with interactions in the examination of wildfire-environment data (Woolford et al., 2010)."

[150] re: the argument for excluding curing from the RF model – does this logic not also extend to the GAM? »> That was included in error from a prior version of the manuscript. It has been deleted.

3 Results [160-170] if you add a description of the predictor variables/datasets in Section 2, you can omit the 'background' info you include here: defining the DMC, explaining derivation of the FFMC, explaining the fact that FWI vars are observed at noon etc. These type of descriptions probably shouldn't appear in a results section. »> Agreed. We added a brief and relevant description of the fire weather metrics in the methods section: "A 3-km grid of daily basic surface meteorology at 12:00 (noon) local time (air temperature, humidity, 10-m wind speed, and precipitation sum over prior 24 h) as well as Canadian Fire Weather Index system variables using inverse-distance weighting (Lee et al., 2002) was constructed for every day during 2002-2018. The rasters

constructed use the same surface station data as McElhinny et al., 2020. The primary Fire Weather Index variables used include the Fine Fuel Moisture Code, Initial Spread Index, Duff Moisture Code, and Drought Code (Lawson and Armitage, 2008). The Fine Fuel Moisture Code (FFMC) is a model of moisture content for fine dead vegetation material at the forest floor of a closed-canopy forest. The FFMC utilizes all of the above basic surface meteorology to estimate drying rate with an exponential drying rate (time to loss of 2/3 of moisture content) of 18 hours. It is used here as a proxy for the moisture content of dense matted grass thatch, with relative humidity alone a better proxy for the moisture content (Miller, 2019) and ignition capacity (Beverly and Wotton, 2007) of standing grass. High FFMC values indicate drier conditions, up to a maximum of 101. The Initial Spread Index (ISI) is the product of the FFMCand the square of wind speed and is proportional to the forward spread rate potential for grasslands and other open vegetated fuels (Hirsch, 1996). The Duff Moisture represents the moisture content of a forest organic soil layer as estimated by a simple precipitation and evaporation model. It has an exponential drying rate of 12 days, and can be considered a metric of the bi-weekly soil moisture budget. Similarly, the Drought Code is a vertical water budget model (Miller, 2020) for a soil column with a 100 mm soil water capacity (similarly, larger values indicate drier conditions). In this manner, the Drought Code has been shown to represent variations in surface water levels (Turner, 1972); a simple vertical water balance of precipitation and evaporation controls surface water extent in the prairies of Canada, where water routing to streamflow and groundwater infiltration is limited (Woo and Rowsell, 1993). As such, the Drought Code is a proxy for the extent of saturated soil areas (wetlands and other surface pond water) that when sufficiently dry, increase the continuity of fuels on the landscape."

[176] I would expand slightly here by highlighting what the significant splines show (I don't think you actually do this anywhere in the main body, but you do refer to the DoY criterion in the abstract?) e.g. wildfires are highly likely when: values of DoY < Ìt'130, WS > 30, curing 65-85%. »> We thank you for that helpful comment, and we add the text in the GAM results section: "Day of Year analysis showed that wildfires are 75% or

more of detections for days prior to early May. Wind speeds over 25 km h-1 or curing fractions between 50 and 85% were also indicators of the likelihood of hotspot cluster being a wildfire over 75%."

Figure 5: You should explain panels (a)-(c) in the caption – at the minute you only mention panel (d). e.g. what are the blue lines (confidence intervals?) and black 'dashes' next to the axes (some kind of rug plot/distribution?). » We now clarify in the figure caption that the axis ticks represent the marginal distribution of the data as a rug plot.

Panel (d) of Figure 5 is a table, and so should be presented as such in the main body rather than as a panel of this figure. From Section 2.4 you suggest hour of detection was incorporated as a spline not a linear predictor, but in (d) it is a linear predictor –which is correct? »> We now treat hour of detection as a linear predictor only. Panel (d) in the GAM plot is now Table 1.

Is there a reason why you didn't also include a plot of probability vs. FFMC in Figure 5 (as well as hour of day, if it was included as a spline?) »> We now treat FFMC as a linear predictor (appropriate for the variable at its high end of 80+ as observed here). This is reflected in the new Table 1 which is the GAM results.

As mentioned earlier, DMC and DC are scaled before being used in the model, so this should be stated in the methods. »> DMC and DC are not scaled, but rather we present the odds ratio of these linear predictors.

Why does RH have an asterisk next to it? I would not use this symbol here as you already use asterisks to signify significance in the same table, which is confusing. »> We have now switched over to superscript numbers for all footnotes in the Figure.

[184-201] decision tree results: This section is currently a bit confusing - I suggest it is restructured slightly, and some clarifications added. Firstly, how many fire clusters in total did you analyse here? I was expecting n=143 (113 wildfires + 41 agricultural

fires stated on line 143, minus the 11 DC < 100 fires mentioned later) but adding up the denominators in Figure 6 it appears that n=95. Assuming I am reading Figure 6 correctly, shouldn't these two numbers match? After introducing the regression tree in Figure 6, It might be worth immediately stating the number of fires analysed, and that you removed the 11 low DC wildfires (plus any other filtering you did?) before discussing the specific results shown by the regression tree, so readers don't spend time looking for the 'missing' fires in figure 6. »> Moved the discussion of the low DC wildfires to just before discussing the results, as suggested.

[185] where is the 92 % accuracy figure from? Should this say 97 %? 92 % is not in Figure 6 or Table 1. »> Fixed that typo, thanks for noticing that.

[186] Not sure why you talk about FFMC here – was FFMC actually used in the regression tree model? It doesn't appear in Figure 6. »> Deleted. Included by accident from an earlier version of the manuscript.

[191] similarly, where does the 82 % value come from? not in Figure 6 or Table 1. »> The 82% refers to the model's sensitivity (which is given fractionally as 0.82) in Table 1. We've revised this to be consistent between the text and the table.

[192] I'm not sure about introducing Appendix A here, or actually including it in the paper at all (1) you don't really highlight what it adds to the study and (2) it uses the large fire dataset (>3000 fires) that you haven't really introduced yet. »> We include Appendix A so the reader less familiar with the Canadian Fire Weather Index System is able to visualize the parameter space of surface meteorology past the critical ISI 15 threshold. The larger database it comes from is less important than the visualization of the parameter space shown, hence why we don't emphasize the larger database here, just cite the sample size.

[193-195] I would move the sentences comparing the GAM to the tree model, because you go from talking about just the tree model on line 192-3, to comparing the two models (193-195), and then back to discussing just the tree model [195-201], which is

structurally hard to follow. »> Removed the comparison to the GAM in the text, as that is obvious in Table 1.

[203-215] this is interesting. Did you try including FRP & wind speed in the tree model? Seems like doing so could have added to tree classification skill? »> We did, but neither came out as significant in the model once ISI was introduced.

[217-220] this paragraph (GAM applied to all clusters) feels like it might work better following the other paragraph on the GAM [lines 175-183] »> Moved as suggested.

[218] should this point to what is labelled as Figure 7 (the one with with two panel plots) rather than Figure 8? »>Revised as suggested.

Figures 7, 8 and 9: These are interesting figures, but you do not have much on them in either the results or discussion section (and in the case of figure 8, the 'avg. no. days per year figure', I don't think you mention this figure at all!). Some explanation is definitely required, otherwise why are they here? »> Explanation of figures 7-9 added to results.

4 Discussion General comment on discussion: Overall, you make some interesting points here, but several of them feel like they need expanding upon. I feel like you also don't draw much from the 'final' outputs of the study (Figures 7-9) – surely these results warrant discussion? Also, this paper clearly has important implications for operational fire management in grassland/agricultural complexes of Canada (and possibly beyond) – while you do mention this, I think you should try to highlight this aspect further in the discussion. »> More discussion on Figs 7-9 was added in the very last paragraph of the discussion.

[222] this paragraph might go better in the introduction/datasets sections of the paper, as it is effectively a justification of why you chose to examine MODIS rather than other options »> Moved to methods and material section.

[232] "> 7500 fires" this statistic is from which dataset? »> Revised to "In all likelihood,

many of the roughly 7 500 wildfire hotspot clusters classified by the GAM over 17 years…"

[244-252] I'm not sure what the key point you are trying to highlight here is, so this probably needs clarifying. I think your main point is that FFMC is a reasonable proxy for grassland moisture content/fire occurrence in the study area? If this is the case, it is interesting to me that (1) FFMC is not significant in the GAM and (2) FFMC is not included in the decision tree model (Fig 6), and this observation might merit further discussion here. »> We've modified this section to remove the focus on the FFMC (which underlies the ISI metric used in the decision tree), and revised this as: "Both the GAM as well as the classification tree point the combination of critically dry fuel and wind as the drivers of wildfire occurrence in the region. In the GAM model, both low RH (as a proxy for standing grass moisture), alongside indicators of bi-weekly (DMC) to monthly (DC) moisture deficit are significant in predicting wildfire occurrence as linear predictors, with a wind threshold in the range of 30 km h-1. In the Canadian Fire Weather Index System, fine fuel moisture (mostly driven by low RH) is combined with wind speeds to calculate the Initial Spread Index as a single heuristic (Appendix A), and thus comes out as the strongest indicator of wildfire. Lindley et al., 2011 found no such moisture deficit as a driver of wildfire occurrence, and instead found that RH alone below 25 % and particularly below 20 % as responsible for most grassland wildfires in west Texas. In our study region, RH alone however is not an ideal proxy for fuel moisture across the wide range of air temperatures found in the region during wildfire, as RH alone does not account for variable vapour pressure deficit at different temperatures (Srock et al., 2018) that drives the equilibrium moisture content of standing grasses (Miller, 2019). Moreover, the extensive shallow water bodies in the region may contribute during periods of higher moisture surplus (i.e. low DMC and DC) to a fragmentation of fuel continuity, similar to the function of larger lakes to the north in Canada (Nielsen et al., 2016). "

[261-266] Interesting observation, and this makes intuitive sense because managed

fires that escape and become wildfires are probably usually the ones that reach the suppression limit. You should probably expand on this slightly though: (1) you could justifiably highlight that this adds to the validity of your work, as you have derived thresholds from a 'top down' RS/modelling approach that agree well with physical, bottom up observations of fire behaviour. (2) maybe you draw this out further? e.g. what might this finding have any applied fire management implications? »> We don't here to to over-extend this analysis, but your point is well taken, and we've added the text to the end of that paragraph: "This correspondence of our remotely sensed records (confirmed by fire reports solely of date and time, not of reported fire behaviour) and the operational models in the Canadian Forest Fire Danger Rating System lends confidence to the application of our approach in public safety and awareness messaging."

[268-276] You highlight an important point - that grasslands are increasing, and likely to keep doing so under climate change and current agricultural conversion trends. But you do not then use these points to highlight the importance of the work you have done here, and that it will be increasingly important in future – I think you should definitely emphasise this! »> Revised to include: "The expansion of grasslands and agriculture into currently forested areas will substantially change the fire regime in these areas, highlighting the importance of understanding the current grassland and agriculture area fire regime. Understanding how fire regimes could change with climate change will help fire managers make long term fire management decisions."

5 Conclusion [283] maybe rephrase to say "a noon ISI threshold of > 15 was the most powerful threshold for discriminating wildfires from agricultural fires, while grass curing. . ." »>Revised as suggested.

Supplementary materials [8] do you mean UTC rather than UTM date and time? »>Revised as suggested.

[39] How were each variable aggregated by fire cluster? E.g. FRP average vs max might be important to know. . . »> Revised to: "An attribute was merged by max value or

mean, for each hotspot cluster. FFMC, DMC, DC, ISI, BUI, FWI, precipitation, relative humidity, wind speed, temperature, hour of detection, percent cured and NDVI were merged by mean. HFI and FRP were merged by max values."

[40] the 5% buffer you describe here, is this the same buffer you indicate in eq S2, or is this an additional buffer? »> Revised to: "Five percent was added to the buffer radius to fix this."

[117-122] this paragraph contains useful detail justification on the number of clusters you used. I would integrate at least some of this information into the main body, as it is important. »> We appreciate the comment, but feel the details on the clustering methodology is too detailed for the main text, and we would like to keep it in the appendix,

Technical corrections Figures: Figure numbers are often incorrect in captions, and in places throughout the text. Please review and amend. Also consider generally expanding figure captions to include more information on the features of the figures or datasets used etc (see specific comments on figures where I believe these could be improved). »> Figure captions were expanded and figure numbers corrected.

[60] consider deleting ". . .despite higher spread rates: : :". Probably adds to an unnecessarily long sentence »>Revised as suggested

[87] "..northern fringe of agriculture.." - not sure if this applies to both areas (i.e. the 'main' southern Prairie area and the distinct northern agri-forest area?) or just the main southern one, please clarify »>Revised to: ". . . At the fringe of agriculture. . ."

[96] "agencies" should have an apostrophe? »>Revised as suggested.

[Figure 4] is labelled as figure 2. You refer to panel letters (a, b etc) in the text but they are missing from the figure. I think this shows results for fire clusters, not MODIS pixel detections – make this clear in the caption and text. Also, the 'Day of year' panel only extends slightly beyond DoY 300 – is this intentional? (maybe there is never fire

after this date?) »> This is the default axis limits in the GAM plotting function in R. The minimum value of the day of year field is 98, and the maximum is 298.

[150] is 'Additionally' a better word choice here than 'Alternately' as you build both models? »> Revised as suggested.

[155] I think you are referring to fire clusters here – if so, I would make this obvious by saying 'distribution of agricultural vs. wildfire clusters' »> Revised to: "...distribution of agricultural vs wildfire hotspot clusters..."

[173] I would state the median no. of pixels for agricultural fires here for comparison to the median wildfire pixels »> Revised to : "The median number of thermal detection points per wildfire was 2 but as high as 55, in contrast with agricultural fires where the median number of thermal detections is also 2 but the maximum is 6."

[178] should this say "increased rate of wildfire likelihood per integer increase in predictor value"? »> Revised as suggested

[257] I'm not very familiar with the use of odds ratios, so ignore this comment if it has a different technical interpretation - but might this be better phrased as "...results in the increase in the odds of a wildfire over an agricultural fire by 2.45..."? »> yes, we've revised this to : "Relative humidity and DC were found to be significant in the GAM model as linear predictors, with odds ratios (increased likelihood of a fire being classified as a wildfire per integer increase in predictor value) of 0.31 per unit increase in RH, and 1.008 per unit of DC. " It is important in odds ratios to state the increasing likelihood of a detection being a wildfire per unit change of the variable of interest.

[273] I think you want 'exacerbated' rather than 'exasperated' here? »> revised as suggested

---

## Author Comment (AC2) · 25 Aug 2020

Reviewer 2: Overall, I think this paper is trying to advance natural hazards - specifically fire science - in using remote sensing and data science to attribute and predict wildland vs. human caused fire. I would recommend the authors refine the terminology. I look forward to reading a revised version.

General comments: 1. Landsat 8 is not an acronym and should not be capitalised.

[Figure]

»>Fixed as suggested.

Referring to all non-wildland fires as agricultural fires becomes confusing later on, especially when trying to explain how the curing data set was included in the regression tree [much of the agricultural landscape was exempted from the curing assessment because < 40% open fuels]. »> We now clarify in the methods that the trees are uncommon in the region outside of shelterbelts: "Within the agricultural ecumene, the vast majority of the region constitutes open fuels (Figure 1), and little tree cover exists outside of shelter belt plantations which exist as single rows of trees (Piwowar et al., 2016). "

The term 'responsible use of fire' is used to encompass a large amount of human-caused burning.    Is this a legal or statute-based definition?    This is not a common term in fire science.    Also, burning of crop residues is not necessarily considered an appropriate thing for this ecosystem.    The Province of Alberta has shifted to no-burn management of crop residues, treating burning as a last resort:    https://open.alberta.ca/dataset/dd5ca66a-09f6-4aeb-8bb9-                        21babed92780/resource/3b67de8e-7377-406c-94d7-25f3efaee710/download/mar2017-unharvested-crops-fs.pdf » We adopt a terminology similar to Lewis et al 2018, where "use of fire" is specific to the low-intensity application of fire in an informal context by community members, not in a formal prescribed fire context. This isn't a definition based on legal statute. As we discuss later in the paper, burning of post-harvest flax residue may be in part responsible for higher fire activity in the eastern portion of our study region.

Why was 2002 (Terra only) MODIS active fire product included when the combined (Aqua and Terra) MODIS active fire product is available starting in 2003? How were these differences in number of detections accounted for when determining the clusters? Was the 2002 Terra-only MODIS active fire useful? »> Only 3 of 140 hotspot clusters were from 2002, so we did not go to the effort of normalizing the lower detection rate of having only one MODIS instrument. For the density analysis across the

landscape, since we had autumn 2002 included and the density data in Figure 6 is an average over the 2002-2018 period, we similarly did not normalize for such a small effect.

Paragraph 265: The thesis statement of this paragraph may need to be re-written " The thresholds at which agricultural fire detections are overtaken by wildfires occurs at fire intensity thresholds that correspond to the limits of ground-based wildfire suppression." Is this a result or a qualitative observation or an assumption that fits into the description of the CFFDRS is the following sentence? Please consider re-phrasing this paragraph. » Rephrased to: "The thresholds shown here in the classification tree and GAM models correspond to modelled fire intensity conditions at the upper limits of ground-based wildfire suppression."

I do not understand how this fits into the study or the findings. Perhaps, again, it is an issue with referring to grass fires as agricultural fires. This reads as the CFFDRS for native grasslands. Is that correct? »> We now clarify the relationship between agricultural debris and fire behaviour models in Canada: "The grass fire spread model in the Canadian Forest Fire Danger Rating System utilizes Australian experimental grass fire data that has been shown to approximate fire behaviour in wheat crops, with the matted (or cut) grass model approximating spring (cured) post-harvest debris (Cruz et al., 2020)."

Is the last paragraph in the discussion section implying increasing agricultural fires with climate change? Did this study find increasing agricultural fires? And if so, in grass-lands or croplands? »> We now clarify that there has been no observed trends in fire activity in the region, though wildfire activity is expected to increase in the surrounding forest regions: "In addition to this likely grassland and cropland expansion, projections of increasingly common critical fire weather conditions (Wang et al., 2015) is likely to shift the fire regime to one of more open fuel burning. However, no change in the rate of fire detections (undifferentiated between wildfires and agricultural burning) has been detected between 1981-2000(Riaño et al., 2007) nor 1998-2015 (Andela et al., 2017)."

---

## Author Comment (AC3) · 25 Aug 2020

We thank the two reviewers for their careful reading of the paper substance as well as layout. We have glady produced a revised manuscripts that satisfies the questions and concerns of both reviewers. We feel the revised manuscript adequately addresses all of their questions and concerns while at the same being a far improved manuscript.